# Integrative determination of atomic structure of mutant huntingtin exon 1 fibrils implicated in Huntington disease

Mahdi Bagherpoor Helabad [1,2,3], Irina Matlahov[4], Raj Kumar[5], Jan O. Daldrop[6], Greeshma Jain [4], Markus Weingarth [5], Patrick C. A. van der Wel [4] ✉ & Markus S. Miettinen [1,6,7,8] ✉

Neurodegeneration in Huntington's disease (HD) is accompanied by the aggregation of fragments of the mutant huntingtin protein, a biomarker of disease progression. A particular pathogenic role has been attributed to the aggregation-prone huntingtin exon 1 (HTTex1), generated by aberrant splicing or proteolysis, and containing the expanded polyglutamine (polyQ) segment. Unlike amyloid fibrils from Parkinson's and Alzheimer's diseases, the atomic-level structure of HTTex1 fibrils has remained unknown, limiting diagnostic and treatment efforts. We present and analyze the structure of fibrils formed by polyQ peptides and polyQ-expanded HTTex1 in vitro. Atomic-resolution perspectives are enabled by an integrative analysis and unrestrained all-atom molecular dynamics (MD) simulations incorporating experimental data from electron microscopy (EM), solid-state NMR, and other techniques. Alongside the use of prior data, we report magic angle spinning NMR studies of glutamine residues of the polyQ fibril core and surface, distinguished via hydrogen-deuterium exchange (HDX). Our study provides a molecular understanding of the structure of the core as well as surface of aggregated HTTex1, including the fuzzy coat and polyQ–water interface. The obtained data are discussed in context of their implications for understanding the detection of such aggregates (diagnostics) as well as known biological properties of the fibrils.

Huntington's disease (HD) is one of a family of incurable neurological genetic disorders resulting from an aberrant expansion of a CAG trinucleotide repeat in a disease-specific gene[1,2]. In HD, the mutation impacts the huntingtin (HTT) protein, which ends up with an expanded polyglutamine (polyQ) tract in its first exon (HTTex1, Fig. 1A)[3]. In HD patients and model animals, mutant HTT fragments matching HTTex1 form inclusions in brain areas affected by

neurodegeneration. In cells, $\mu$m-sized inclusions contain fibrillar protein aggregates with widths of several nanometers, as seen by cryogenic electron tomography (cryo-ET)[4–6]. Fibrillar HTTex1 has all the common features of amyloid fibrils found in diseases such as Parkinson's and Alzheimer's disorders: highly stable protein deposits built around extended intermolecular $\beta$-sheets forming a characteristic cross-$\beta$ architecture[7–9]. The protein fibrils are of interest due to

[1]Department of Theory and Bio-Systems, Max Planck Institute of Colloids and Interfaces, 14476 Potsdam, Germany. [2]Institute for Drug Discovery, Leipzig University Medical Center, 04103 Leipzig, Germany. [3]Institute of Chemistry, Martin Luther-University Halle-Wittenberg, 06120 Halle (Saale), Germany. [4]Zernike Institute for Advanced Materials, University of Groningen, 9747 AG Groningen, The Netherlands. [5]NMR Spectroscopy, Bijvoet Centre for Biomolecular Research, Department of Chemistry, University of Utrecht, 3584 CH Utrecht, The Netherlands. [6]Fachbereich Physik, Freie Universität Berlin, 14195 Berlin, Germany. [7]Department of Chemistry, University of Bergen, 5007 Bergen, Norway. [8]Computational Biology Unit, Department of Informatics, University of Bergen, 5008 Bergen, Norway. ✉e-mail: p.c.a.van.der.wel@rug.nl; markus.miettinen@mpikg.mpg.de

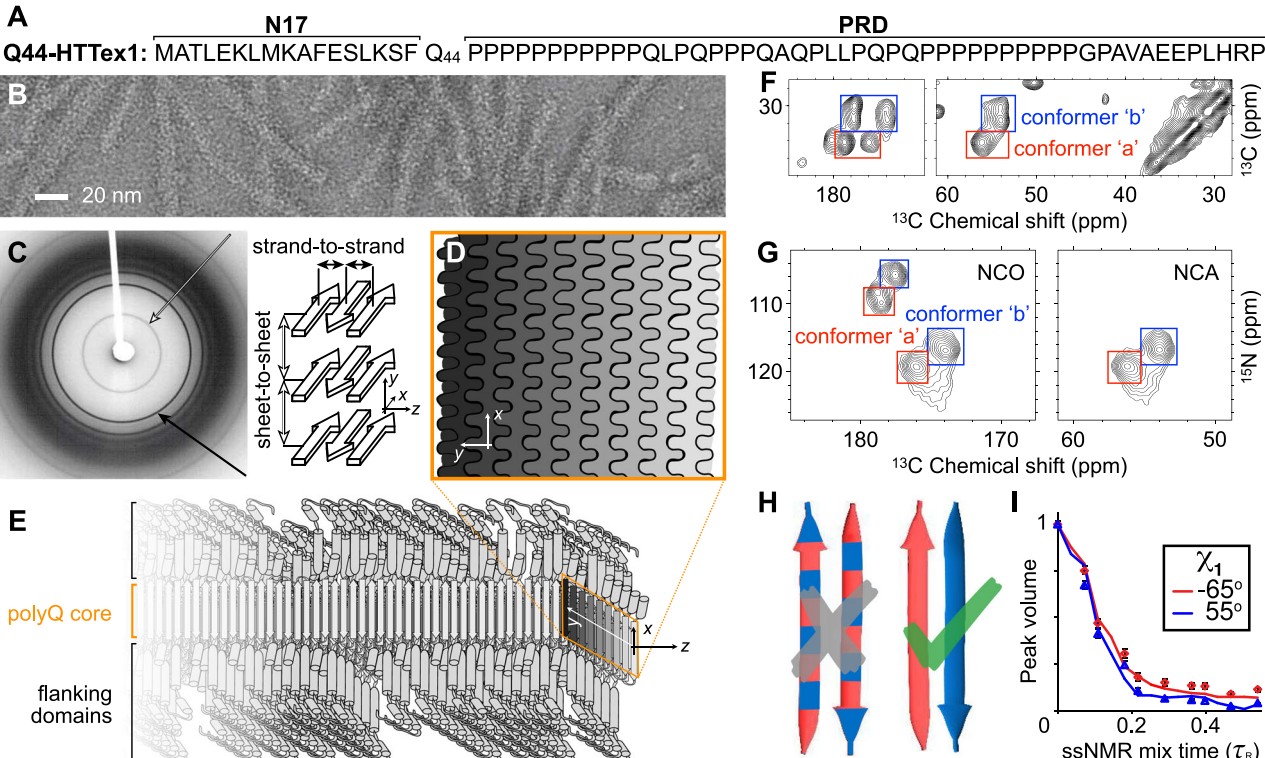

**Fig. 1 | Select structural data on HTT Exon 1 aggregates. A** Sequence of Q44-HTTex1. N- and C-terminal flanking domains marked as N17 and PRD, respectively. **B** Negatively-stained TEM of Q44-HTTex1 aggregates formed in vitro; average fiber width ~6.5 nm. **C** X-ray fiber diffraction of K₂Q₃₁K₂ fibrils, detecting the cross-β reflections of the amyloid core that represent the inter-strand and inter-sheet distances shown on the right. **D** Cross-section of the (6–7-nm wide) block-like core showing the layering and interdigitation of (differently shaded) β-sheets. **E** Schematic model of a Q44-HTTex1 fibril, showing the N17 and PRD flanking domains outside the polyQ core. **F** 2D ¹³C–¹³C ssNMR spectrum of ¹³C,¹⁵N-labeled Q44-HTTex1 fibrils showing backbone and sidechain cross-peaks of the dominant signals of the polyQ fibril core. Peaks for Cα–CX correlations of two distinct "a" and "b" Gln conformers are marked with red and blue boxes, respectively. **G** 2D NCO and NCA ssNMR spectra of Q44-HTTex1 fibrils, with the conformers "a" and "b" again marked. **H** Pairs of antiparallel β-strands, schematically showing two previously proposed arrangements of the "a" (red) and "b" (blue) types of Gln. Given the ssNMR data (panels **F**–**G**), only the right one fits the experimental data. See also Supplementary Fig. 1. **I** ssNMR measurements sensitive of the side-chain dihedral angle χ₁, with simulated curves for "a" (red) and "b" (blue) types of Gln. Error bars indicate experimental error in measured peak volumes due to noise, based on analysis of a single sample (n = 1). See also ref. 8. Panels (**B**, **C**) and (**I**) adapted with permission from ref. 8, (**D**, **E**) from ref. 41, and (**G**) from ref. 53. Source Data file for panel (**F**) provided in ref. 85.

their potential involvement in pathogenic and neurotoxic processes −but also as biomarkers of disease onset and progression. Consequently, there is a need to resolve their atomic structures to facilitate the design of inhibitors, modulators, and diagnostic tools. For instance, a better understanding of HTTex1 fibril structure is essential to the development of ligands used to detect HTTex1 fibril formation in vivo[10].

Recent years have seen important progress in amyloid structure determination through application of cryogenic electron microscopy (cryo-EM) and solid-state NMR spectroscopy (ssNMR)[7,11,12]. Breakthrough studies have produced structures of amyloid fibrils formed by tau, amyloid-β, and α-synuclein, associated with Alzheimer's, Parkinson's, and other amyloid disorders[12]. The polyQ-based amyloid fibrils from HD and other CAG repeat disorders[2], however, have proved more challenging, and still lack truly atomic-level structures or structural models, limiting progress in mechanistic and diagnostic research. The challenges in studying these proteins stem in part from their atypical amyloidogenic motif, formed by repetitive sequences. The aggregation propensity of polyQ proteins correlates strongly with their polyQ tract length, as determined by the inherited mutation, which inversely correlates with the age of onset[2,13]. In their native state, polyQ segments are typically considered to lack a stable secondary structure, forming an intrinsically disordered region. This is also true for the polyQ-containing segment of HTT (HTTex1), which is invisible in cryo-EM of the unaggregated full-length human HTT protein[14,15]. However,

in HD, N-terminal fragments of polyQ-expanded HTT are generated from protease activity and aberrant splicing, enabling their subsequent aggregation into fibrils and ultimately cellular inclusions[16–18]. This has lead to considerable interest in studies of HTTex1 aggregates as an important mechanistic factor in HD pathogenic mechanisms.

In in vitro structural studies, HTTex1 fibrils (Fig. 1B) featured a highly ordered amyloid core, formed by the polyQ segment, surrounded by flexible non-amyloid flanking regions[6,19–23]. Prior fiber X-ray diffraction studies showed the core itself to comprise anti-parallel β-sheets in a cross-β pattern (Fig. 1C)[24–27]. These experiments spurred proposals of various possible fibril models[24–27]. However, such models were typically presented as illustrative rather than an atomic-level analysis, due to the lack of availability of sufficient experimental information. Moreover, all early X-ray-based models proved inconsistent with later work[8,28–30] that revealed important new structural data via combinations of cryo-EM, ssNMR, and other techniques[8,19–22,30–36]. The current study builds on valuable data from various prior experimental studies, and provides additional (NMR) experimental data, to perform an integrative structural analysis of polyQ amyloid and HTTex1 fibrils. A few recent studies are of particular interest. Firstly, an integration of ssNMR, X-ray, and EM measurements permitted the manual assembly of a schematic fibril architecture in which the polyQ segments form a block-like fibril core (Fig. 1D–E)[8,20,28,29]. Subsequently, a cryo-EM study of HTTex1 fibrils[23] produced a medium-resolution density map, which−although it lacked

the detail for a de-novo atomistic structure—was interpreted to be consistent with the abovementioned model architecture: the amyloid core has a block-like structure assembled from multiple layers of tightly packed $\beta$-sheets (as schematically shown in Fig. 1D–E). Notably, this type of architecture is distinct from more typical amyloid fibrils (Supplementary Fig. 1). The apparent success of such a unified (but as-yet qualitative) model underlines the need for a modern integrative-structural-biology approach[37] to fully describe these fibrils that have much in common with other amyloids, but also a number of truly unique features.

Here we employ rigorous physics-based molecular modeling to integrate the collective experimental knowledge into atomic-resolution polyQ and HTTex1 fibril structures that are fully consistent with all the key experimental data, including sheet-to-sheet distances from fiber X-ray diffraction, fiber dimensions from EM, and structural constraints from ssNMR. We combine previously reported experimental data with additional experimental measurements, with the latter in particular examining the fibril core and surface features that were previously not observed. The obtained structures are analyzed in the context of known structural features of HTTex1 amyloid fibrils, allowing a detailed explanation of their conformational and spectroscopic characteristics. Moreover, based on the disposition and accessibility of key residues and segments, the molecular architecture of these HD-related fibrils explains biological properties, such as the (in)ability for post-aggregation polyubiquitination and degradation.

## Results and discussion
### Architecture of the internal polyQ fibril core
We started the integrative structure determination of HTTex1 fibril from its polyQ amyloid core. For this, we first review previously reported experimental data that inform our modeling approach. Most experimental studies, in particular recent ssNMR studies (Supplementary Fig. 1A), argue for a shared molecular architecture common to polyQ-containing peptide and protein fibrils[36,38]. Various techniques show that polyQ amyloid features antiparallel $\beta$-sheets (Fig. 1E)[26,27,30,33,39], in contrast to the parallel in-register fiber architecture of many other amyloid fibers[11,40]. Another distinct feature of polyQ amyloid is that it forms long $\beta$-strands with few turns (Fig. 1F; Supplementary Fig. 1B)[8,23,27,29,41], unlike the majority of other amyloid structures that have short $\beta$-strands connected by turns or loops[11]. The polyQ fiber core is multiple nanometers wide, devoid of water molecules, and forms a block-like structure featuring a stacking of less than ten $\beta$-sheets (Fig. 1D)[23,27–29]. Its cross-section must contain polyQ segments from multiple protein monomers (schematically illustrated in Fig. 1D–E)[41]. A large proportion of the Gln residues is buried, far away from the solvent, in a repetitive and pseudo-symmetric context. This ordered and repetitive nature is demonstrated by high-quality ssNMR spectra, with few and relatively narrow peaks (Fig. 1F–G; Supplementary Fig. 1)[8,20,22,28,29,32,41]. These ssNMR studies consistently show a characteristic spectroscopic fingerprint that identifies two dominant Gln residue conformations—see the two sets of peaks from so-called "a" and "b" conformers (red and blue boxes) present at roughly equal intensities in the 2D ssNMR spectra of Q44-HTTex1 fibrils (Fig. 1F–G; Supplementary Fig. 1). The two Gln conformers reflect two types of $\beta$-strand structures that co-assemble into a single antiparallel $\beta$-sheet (Fig. 1H; more below)[8,22,29].

Integrating these known experimental data, we concluded that the internal polyQ core structure can be captured by an infinitely-repeating eight-Gln unit that features antiparallel $\beta$-sheets (Fig. 2A). This construction has translational symmetry and lacks the twisting seen in many amyloid fibrils, since EM analyses of our and others' HTTex1 fibrils lack strong signs of twisting[23]. The repeating unit cell (Fig. 2A) fulfills several a priori requirements: First, its peptide chain segments are two residues long, the minimum needed to capture the odd/even side-chain alternation in a $\beta$-strand. Given that ssNMR shows

only two ssNMR-detected conformers, in long strands (Supplementary Fig. 1; see refs. 8,22,29), two residues should be sufficient to capture the structural variation within this amyloid core. Second, the unit cell contains two $\beta$-strands (that can differ in conformation), the minimum needed to describe an antiparallel $\beta$-sheet. This contrasts with parallel in-register $\beta$-sheets that could be represented with a single repeating $\beta$-strand. Third, the unit cell contains a stacking of two $\beta$-sheets, to permit the probing of distinct sheet–sheet dispositions. Fourth, certain ranges of side-chain dihedral angles ($\chi_1, \chi_2, \chi_3$) are required in order to fulfill a crucial feature of the interdigitating polyQ amyloid:[8,26] the side-chain–side-chain hydrogen bonding between the strands (see Fig. 2A and Supplementary Fig. 2). Thus, the $\chi_2$ dihedral must be close to 180°, as also known from ssNMR and Raman experiments[8,42]. This minimal model (Fig. 2A) fulfills the above-noted experimental constraints, and still yields a library of roughly 1 000 distinct architectures (see Methods for details).

A key additional consideration is that ssNMR has unambiguously shown sequential residues within each $\beta$-strand to have the same backbone conformations (based on identical chemical shifts; Supplementary Fig. 1B)[8,22,36]. In summary, the "a" and "b" conformers strictly occupy distinct strands, within which uniform backbone and $\chi_1$ dihedral angles are found (Fig. 1H). Moreover, these two distinct $\beta$-strand types differ in their $\psi$ and $\chi_1$ dihedral angles (Fig. 1I; Supplementary Fig. 1D, E)[8]. Thus, these additional constraints can be applied to the initial library of architectures. After applying this additional filtering, 30 distinct unit cell architectures still remained possible (Supplementary Fig. 3). To investigate the structural stability of these 30 models, we carried out all-atom MD simulations in a fully periodic infinite-core setting; see Methods for details. Only two of the candidate models proved stable throughout the simulations (Fig. 2B and Supplementary Fig. 4).

### Zippers and ladders within the polyQ core
Notably, these two stable 3D lattices (denoted *M1* and *M2*) capture, but also refine, known features of polyQ amyloid structure. Experimental constraints offered by previously reported ssNMR dihedral angle measurements are open to more than one interpretation[8,43]. Previously, this inherent ambiguity yielded qualitative models useful for illustrative purposes. Here, we have rigorously narrowed down the viable and physically plausible models, obtaining only two models that identify specific narrow regions within the equivocal dihedral angle space (Fig. 2D). Interestingly, unlike all the 28 unstable models, *M1* and *M2* display no Gln residues with $\chi_3 \approx 180°$ (Supplementary Fig. 6). In both models, the antiparallel $\beta$-sheet harbors strand-specific Gln structures occupying the side chain rotamers known as pt20° and mt-30°, as defined by ref. 44. These conformations enable hydrogen bonding interactions between the side chains of the Gln conformers, in addition to typical strand-to-strand backbone hydrogen bonds (Fig. 2E). This hydrogen bonding pattern is reminiscent of glutamine (or asparagine) ladders found in other amyloids[45,46]. However, a key difference is that here they occur in an antiparallel $\beta$-sheet, whereas most prior cases are parallel. A recent paper[46] provided a detailed analysis of this type of ladder in HET-s fibrils, noting the relevance of $^1H$ shifts of the side chain amide nitrogens. Their hydrogen-bond strength has a noted effect on these $^1H$ shifts[47,48], resulting in a characteristic pattern for glutamine side chains H-bonded in such ladders: a big difference between their $H_Z$ and $H_E$ shifts (nomenclature in Fig. 2F), with the former reflecting hydrogens involved in strong H-bonds along the fiber axis. Thus, we performed 2D NMR to detect and assign these protons in the polyQ amyloid core (Fig. 2G, Supplementary Fig. 7). These spectra show the backbone N–H signals, as well as peaks for the side chain $NH_2$ groups of the "a" and "b" Gln conformers. Strikingly, these polyQ side chains feature widely

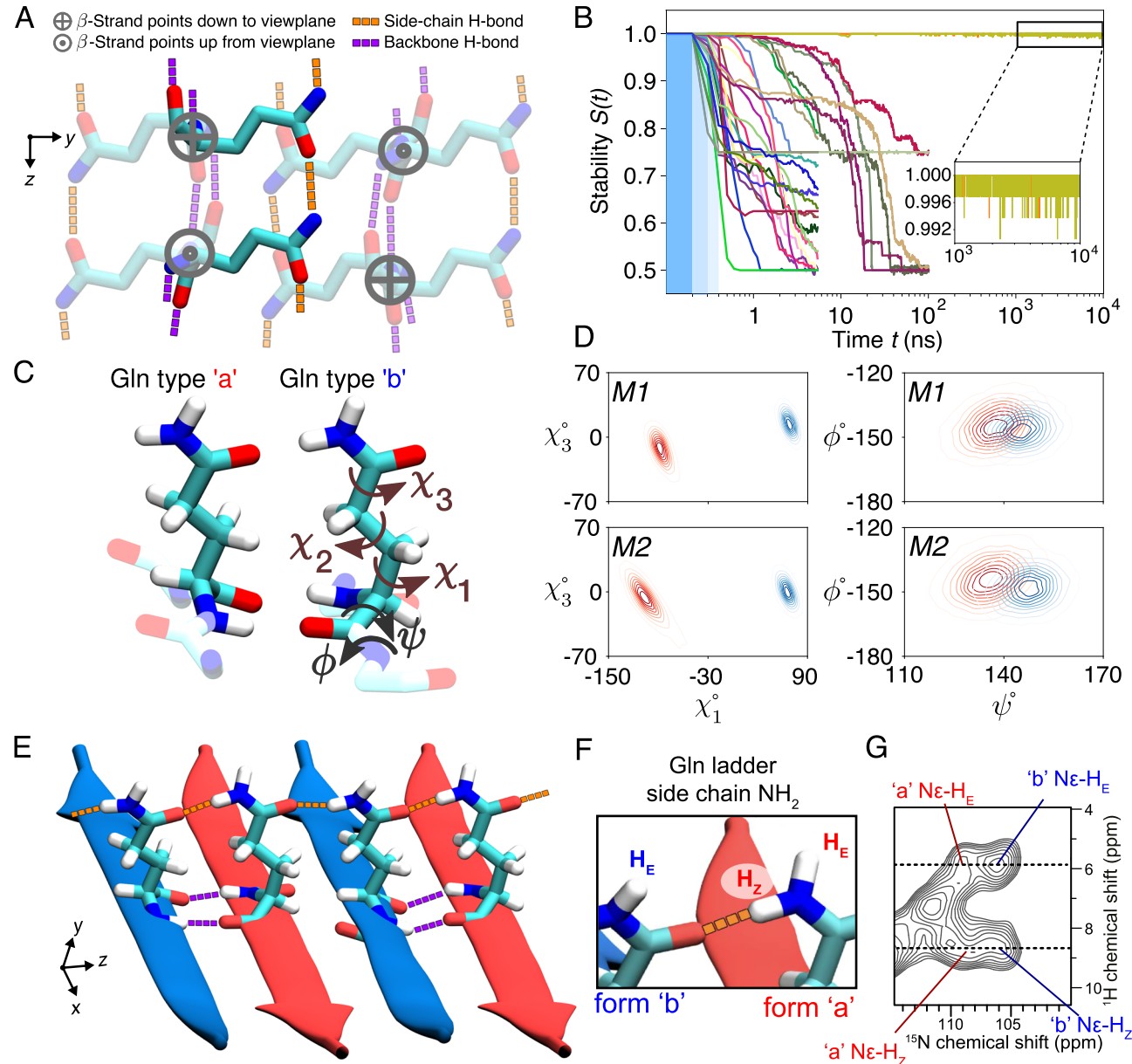

**Fig. 2 | Glutamine zippers and ladders within the polyQ amyloid core. A** The eight-Gln building block used to generate polyQ core candidates. Six residues are shown faded out to bring the two non-faded in focus: these two are on adjacent $\beta$-strands of an antiparallel $\beta$-sheet, stabilized by backbone hydrogen bonds (purple), and continuous chains of side-chain hydrogen bonds (orange). The latter are crucial for packing the polar glutamines into the waterless amyloid core. When generating the core candidates, all the $\chi_1$ and $\chi_3$ dihedral angles were independently rotated to explore all potential hydrogen bond networks. **B** Stabilities (see Eq. (1) in "Methods") of the 30 experimentally-feasible polyQ core candidates (represented by color-coded lines) as a function of MD simulation time. The three blue-shaded sections indicate the gentle protocol chosen for initiating simulations from the energy-minimized ideal structures: Decreasing position restraints (of 1 000, 500, and 100 kJ/mol/nm²) over three consecutive 100-ps periods led to the unrestrained simulation. Notably, only two candidates maintain stability throughout the 10-μs MD simulations; we denote these *M1* and *M2*. **C** Atomic-level structures of the type "a" and "b" Glns in *M2*. Gln dihedral angle names shown on "b". **D** The side-chain $\chi_1$–$\chi_3$ (left panels) and backbone $\psi$–$\phi$ (right panels) dihedral angle distributions of conformers "a" (red) and "b" (blue) for the final models *M1* (top) and *M2* (bottom). **E** Illustration of the inter-side-chain hydrogen-bond ladders (orange) in *M2*. **F** Nomenclature for N$\epsilon$ protons involved in the H-bonding along the ladder (H$_Z$) or orthogonal to it (H$_E$)[46]. **G** 2D $^1$H–$^{15}$N MAS NMR spectrum on HTTex1 fibrils (see also Supplementary Fig. 7). Four cross-peaks are marked for the Gln side-chain NH$_2$ protons of the "a" and "b" conformers that form the core. The dashed lines mark the $^1$H shifts for the H$_Z$ and H$_E$ protons, showing that they are identical for the two conformers. Panels (**B**–**E**) from the Amber14SB[86] force field; for OPLSAA/M[87] and CHARMM36m[88] see Supplementary Figs. 4, 5. Source Data files for panels (**B**, **D**, and **G**) provided in ref. 85.

separated H$_Z$ (~8.2 ppm) and H$_E$ (~5.5 ppm) $^1$H shifts. This large H$_Z$–H$_E$ shift difference actually exceeds that seen in the HET-s fibril[46]. This suggests a particularly strong H-bond interaction in polyQ amyloids, given that the $^1$H shift values for H$_Z$ are connected to the H-bonding distance[46–48]. The resolution of the HET-s fibril structure does not permit a direct translation to an atomic distance, but analysis of Gln ladders in other amyloids suggests side chain oxygen–nitrogen (heavy atom) distances of 2.7 to

2.9 Å[49]. Our stable structures for polyQ (Supplementary Fig. 8) show side-chain–side-chain H-bond distances (for H$_Z$ H-bonds) that match such values.

A notable feature of our models is that these observed distances are the same for the two "a"/"b" conformers. Residues in the HET-s fibril core show substantial differences in their $^1$H shifts[46], presumably due to differences in local structure and H-bonding distance. In contrast, the H$_Z$ and H$_E$ $^1$H shifts in the polyQ core are indistinguishable

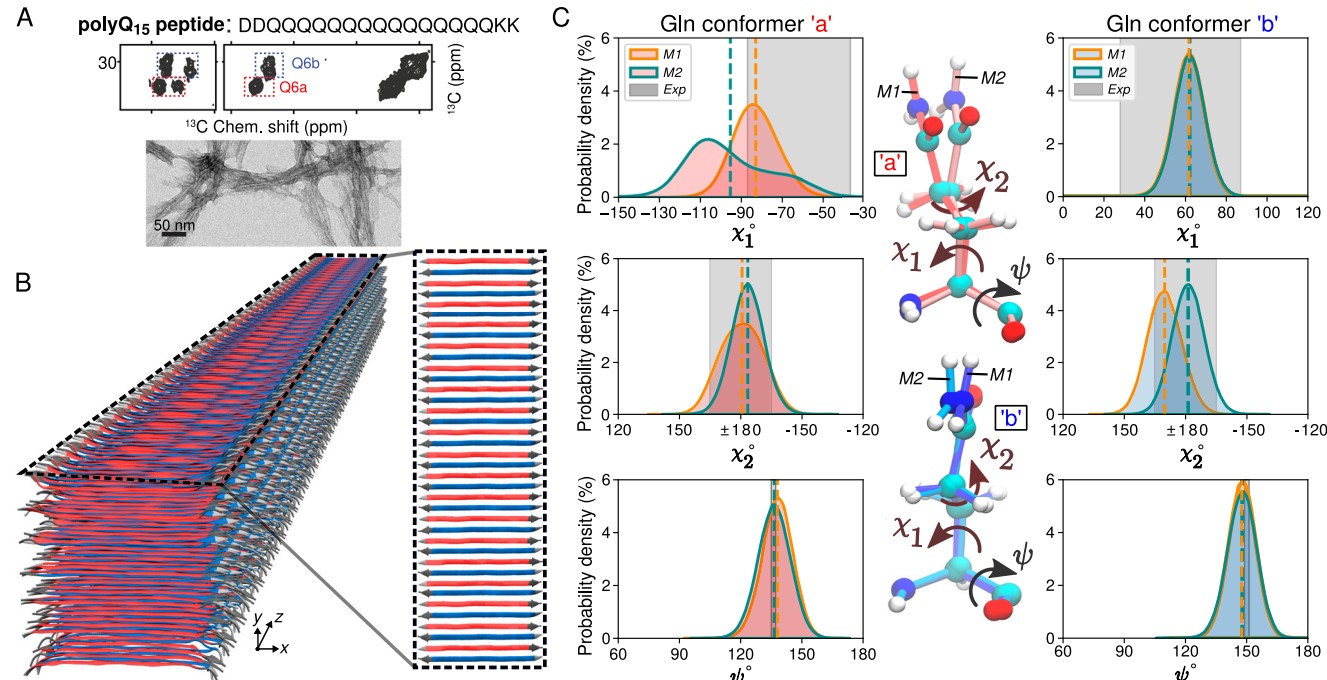

**Fig. 3 | PolyQ peptide fibril structure. A** Amino acid sequence of $D_2Q_{15}K_2$ peptide (top), 2D $^{13}C$–$^{13}C$ ssNMR spectrum (middle) of $D_2Q_{15}K_2$ fibrils with single $^{13}C$-labeled Gln (Q6), and negatively-stained TEM of the peptide aggregates (bottom). The ssNMR data were adapted from ref. 8. Signals from type "a" and "b" Gln conformers are shown in red and blue boxes, respectively. The EM data are representative results from twenty micrographs recorded on a single prepared EM grid. **B** 3D cartoon representation of the structural model of the $D_2Q_{15}K_2$ fibril. The alternation of $\beta$-strands of "a" (red) and "b" (blue) Gln conformers within a single sheet is shown on the right. **C** The $\chi_1$, $\chi_2$, and $\psi$ dihedral angle distributions of "a" (left) and "b" (right) conformers in the context of the $D_2Q_{15}K_2$ fibril for the *M1* (orange) and *M2* (green) models; dashed vertical lines represent mean values. The data were obtained from 1-$\mu$s MD simulations (using the OPLSAA/M[87] force field; for Amber14SB[86] see Supplementary Fig. 10). Gray shading depicts the ssNMR constraints for the dihedral angles. The structures at the center show representative "a" (top) and "b" (bottom) conformers of the *M1* and *M2* models. Source Data files for panel (**C**) provided in ref. 85.

between the two conformers (Supplementary Fig. 7), consistent with the models' indication that the corresponding hydrogen-bonding interactions are alike for the two conformers in the polyQ core. Notably, these observations were not employed as restraints in our modeling setup. The close match between these simulated distances and the provided NMR data on the Gln side chains (in the fibril core) offers experimental support for the validity of our modeling efforts.

## Structure of polyQ$_{15}$ peptide fibrils

The above 3D models capture the internal core of polyQ protein fibrils, but lack the defined width and water-facing surface features of the real polypeptide fibrils. To model a proper experimental system, we modified the infinite cores *M1* and *M2* into models of fibrils formed by the polyQ$_{15}$ peptide (Fig. 3A). These widely studied[8,24,27,50,51] peptides contain a 15-residue polyQ segment, flanked by charged residues to enhance solubility. Experimental analysis[8,29,52] by MAS NMR and X-ray diffraction has shown that atomic conformations in the polyQ$_{15}$ fibril core match those of the disease-relevant HTTex1 fibrils; e.g., the previously published[8] ssNMR signals from Gln within the $D_2Q_{15}K_2$ fibrils (Fig. 3A) and the Q44-HTTex1 fibrils (Fig. 1F) are indistinguishable. Using the *M1* and *M2* models, we constructed $D_2Q_{15}K_2$ fibrils comprising seven antiparallel $\beta$-sheets, consistent with the 5.5–6.5-nm fibril width seen experimentally by TEM (Fig. 3A). We then subjected both polyQ$_{15}$ models to 1-$\mu$s atomistic MD simulations in explicit water; for details, see Methods. Figure 3B shows the *M2* $D_2Q_{15}K_2$ fibril in cartoon representation. The structure of the buried Gln residues within the core remained stable throughout the simulations (Supplementary Fig. 9).

There is a high degree of agreement between the $\chi_1$, $\chi_2$, and $\psi$ dihedral angle distributions observed in MD simulations of our polyQ$_{15}$ models and the ssNMR angle constraints previously reported, based on HCCH ($\chi_1$, $\chi_2$) and NCCN ($\psi$) experiments (Fig. 3C and

Supplementary Fig. 10). Experiments[8] have shown that the "a" and "b" ssNMR signals reflect different $\chi_1$ and $\psi$ values in the two Gln conformers, but a similar $\chi_2$ value. The simulations of both polyQ$_{15}$ models are consistent with these NMR data (Fig. 3C). The largest deviation affects the *M2* model, in terms of its broader distribution of the conformer "a" $\chi_1$ angle, which in part lies outside the shaded region representing the (ambiguous) ssNMR constraint. Experimental studies have shown[8] that the conformer "a" is more dynamic than conformer "b", with additional experimental evidence discussed below. These results are reminiscent of the increased heterogeneity of the "a" conformer in the MD simulations of both models (see also Supplementary Fig. 9). This dynamic difference thus supports our MD results and dynamic averaging effects may help explain the noted (modest) deviation. Also, the sheet-to-sheet and strand-to-strand distances within the models match well with repeat distances from X-ray diffraction (Supplementary Fig. 11). In sum, the MD simulations support the stability of these polyQ amyloid core structures and recapitulate key data from multiple experimental techniques.

## Surface features of polyQ amyloid

Our modeling reveals an important feature of polyQ protein or peptide fibrils, which, so far, has not been seen or studied: the solvent-facing external structure of the polyQ amyloid core. The fiber surface mediates interactions with cellular surroundings but also with any purposefully administered compounds, such as thioflavin-T or amyloid-specific tracers used in positron emission tomography (PET)[10]. In our models, the two outermost $\beta$-sheets exposed to water (top and bottom sheets in Fig. 3B) allow us to compare the surface-exposed Gln residues versus the residues buried in the fibril core. Figure 4A highlights the water-exposed Gln side-chains (green) in the polyQ$_{15}$ fibril model. Figure 4 shows the $\chi_1$–$\chi_2$ and $\chi_3$–$\chi_2$ dihedral angle distributions of the

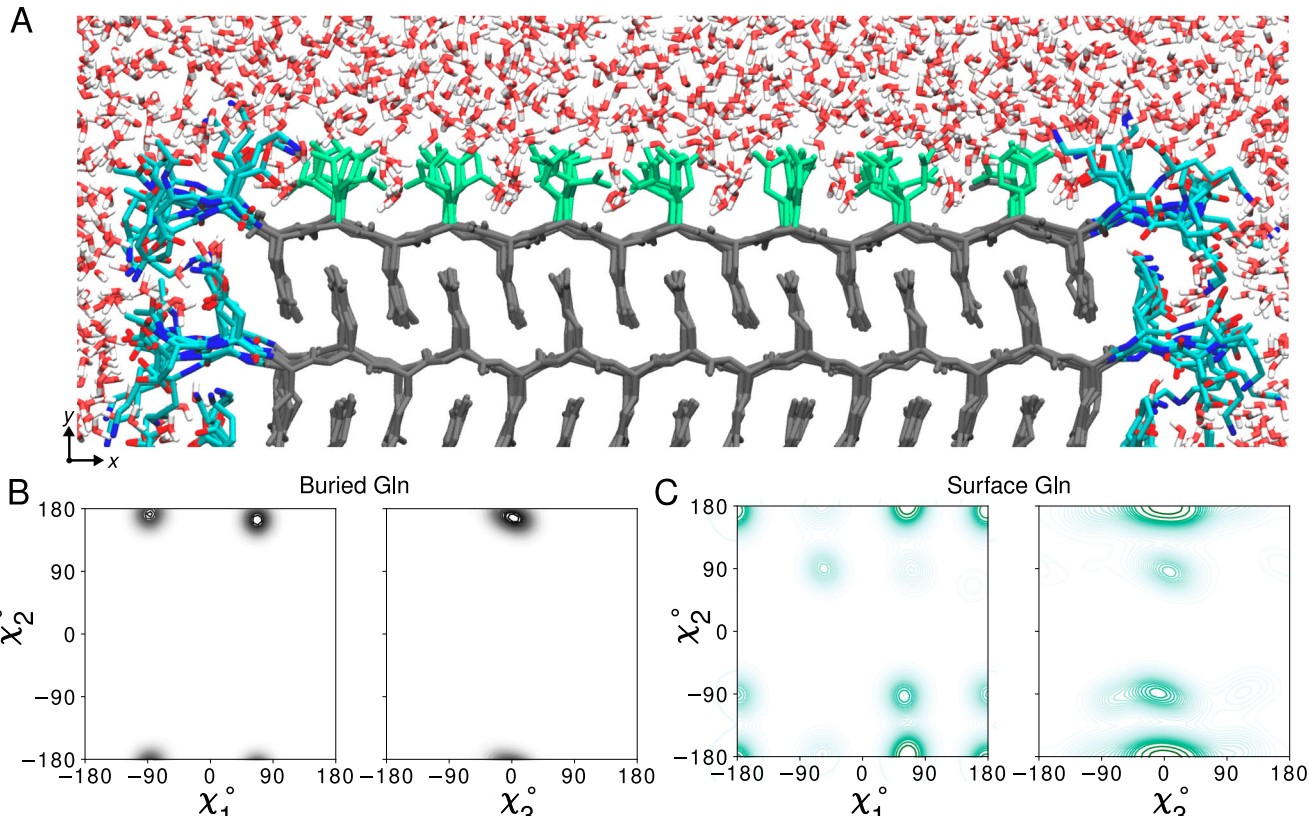

**Fig. 4 | Atomic model of the water-facing surface of polyQ amyloid. A** Atomistic MD snapshot of the $D_2Q_{15}K_2$ peptide fibril's polyQ surface in contact with water. Exposed and buried Gln residues are colored green and gray, respectively. Note how the Gln side-chains internal to the (model *M1*, for *M2* see Supplementary Fig. 12) amyloid core are well-ordered, while the water-facing side-chains display more mobility. **B** Side-chain dihedral angle distributions for the buried Gln residues and (**C**) for the Gln residues on the fibril surface (Amber14SB[86]; for OPLSAA/M[87] see Supplementary Figs. 13, 14). The surface-facing residues show more disorder, but are nonetheless constrained to just few varyingly prominent specific rotamer states. Source Data files for panels (**B**, **C**) provided in ref. 85.

core Gln residues (panel B, gray) and compares them with the surface residues (panel C, green) for model *M1* (see Supplementary Fig. 12 for *M2*). For the core residues, the rotamer states are similar to those discussed above (pt20˚ and mt-30˚). For the surface residues, additional rotameric regions emerge, including Gln rotamers where $\chi_2$ deviates from 180˚, close to rotamer pm0˚[44]. Interestingly, surface residues do not show fully free side-chain motion and retain some of the structural features that determine the molecular profile of the corrugated polyQ amyloid core surface. This observation is highly relevant for efforts to design selective binders of the polyQ amyloid surface.

Until now, few experimental data inform us about these polyQ–water interfaces. A big challenge in understanding the surface-exposed Gln residues is that their NMR signals overlap with those of the core, which are more numerous and therefore dominate the observed signals[53]. Indeed, we (the authors) had assumed the surface and core signals to be more similar than predicted by the MD results (Fig. 4). To evaluate this important feature, we designed and performed experimental studies combining hydrogen–deuterium exchange (HDX) with advanced MAS NMR analysis. The N–H bonds within the fibril core are resistant to H/D exchange, due to their dehydrated nature, stable hydrogen bonding, and lack of solvent access[54]. This feature allows one to differentiate and compare the NMR signatures of the core and surface of polyQ protein fibrils. We produced Q44-HTTex1 fibrils that were aggregated either in regular (protonated) buffer or in deuterated buffer. In such fibrils, one expects the exchangeable amide hydrogens to be either intact or exchanged for deuterium, respectively. By TEM, no deuteration-related effects were noted on the fibril morphology (Supplementary Fig. 15A, B). First,

[1]H-detected MAS NMR analysis of the fully protonated fibrils produced the 2D and 3D [1]H–[15]N spectra shown in Fig. 5A and Supplementary Fig. 15D, featuring primarily signals from the rigid polyQ core. Note that no strong signals are expected from the numerous HTTex1 proline residues, as they lack backbone amide protons. The observed signals match those expected for the glutamine backbone amides ([15]N frequency 115–125 ppm) and the side chains ([15]N frequency near 100–115 ppm). Assignment of the polyQ core signals was based on abovementioned 2D correlation experiments (Supplementary Fig. 7). Next, 2D MAS NMR was performed on the partly HDX-exchanged fibrils after exposure to protonated or deuterated buffer (Fig. 5B–C). One expects to either observe only buried residues, or only exposed residues. These spectra reveal the protonated-core signals to match those of that we already assigned to polyQ amyloid (Supplementary Fig. 7). However, the deuterated-core fibrils gave different peaks, which we attribute to the surface exposed glutamines (Fig. 5C). We will examine their distinct shifts below. As another key indicator of surface-exposure, we also performed relaxation measurements (Supplementary Fig. 15G–J). We observed that the [15]N backbone and side chains in the fibril core displayed the relaxation characteristics of rigid residues. In contrast, the side chain [15]N of the surface-exchangeable sites displayed faster relaxation ([15]N $T_1$ and $T_{1\rho}$). Exchangeable backbone [15]N (on the fibril surface) showed intermediate behavior that more closely resembled the polyQ core, in contrast to the side chains. Here it is important to note that the surface residues are more mobile than their buried counterparts, but that their motion is nonetheless greatly restricted. First, these signals are detected via cross-polarization NMR that fails for flexible residues (such as those in the PRD tail). Second, the relaxation properties are indicative of constrained motion,

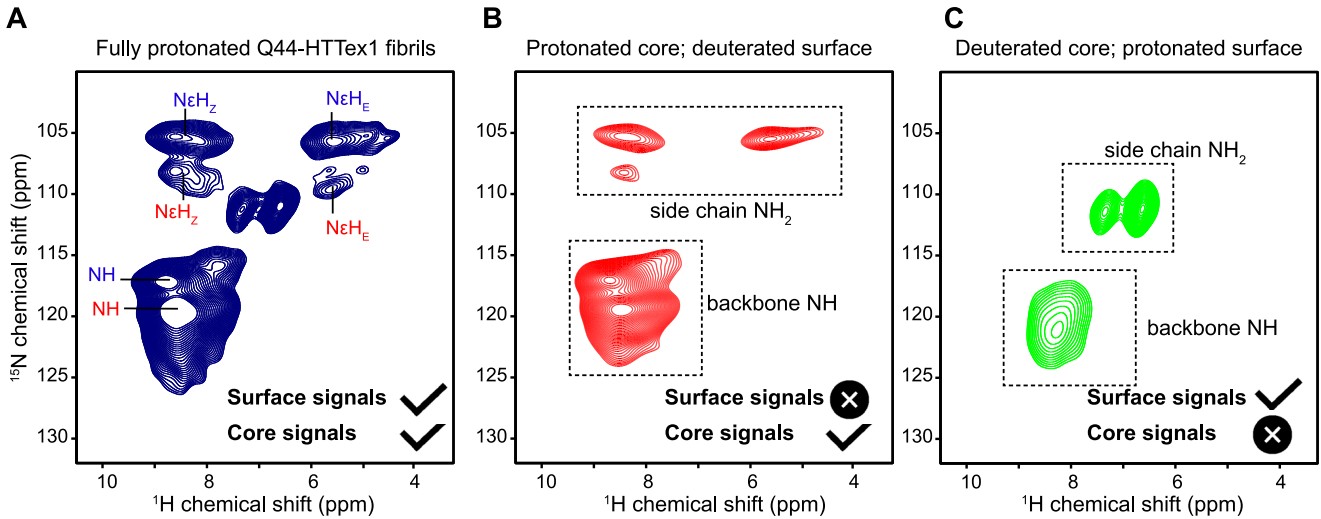

**Fig. 5 | NMR analysis of polyQ core and surface residues based on H–D exchange. A** 2D ^1H-detected ^1H–^15N HETCOR NMR spectrum of fully protonated Q44-HTTex1 fibrils. The peak labels are color-coded based on the conformer type ("a" = red; "b" = blue), corresponding to the amyloid core assignments from Supplementary Fig. 7. Attenuation of peaks from the "a"-conformer side-chains is attributed to different dynamics (see also Supplementary Fig. 15D). **B** Analogous data for surface-deuterated Q44-HTTex1 fibers, which are expected to only show peaks from the fibril core. **C** Analogous 2D spectrum for core-deuterated, surface-detected Q44-HTTex1 fibers, which reveals distinct signals from residues on the polyQ surface. The most dramatic difference is seen for the side chain $NH_2$ group. Measurements at 700 MHz using 60-kHz MAS, at 253 K setpoint temperature. See also Supplementary Fig. 15 for additional data and relaxation measurements. Source Data files provided in ref. 85.

especially considering the relaxation properties of the observed backbone signals.

Thus, we experimentally observed solvent-exposed Gln residues accessible on the polyQ core surface. The chemical shifts of their backbone nitrogens are similar to those of the core residues, suggesting a similarity to the amyloid core backbone structure. However, the side chain ^15N and ^1H shifts are very different from the core. Notably, whilst the characteristic side chain $H_Z$ and $H_E$ shifts can be recognized quite clearly, they are much closer together than the already-discussed core residues. Thus, following the analysis discussed above, this implies the absence of ordered Gln-ladders on the fibril surface. These experimental indicators of dynamics of the side chains on the surface, but more order of the backbones, are highly consistent with the MD showing restricted dynamic disorder on the fibril core surface (Fig. 4C).

## Structure of HD-relevant HTT exon 1 fibrils

As already noted, protein inclusions seen in HD patients and HD model animals incorporate the mutant HTTex1 protein fragment. Until now, no atomistic model of HTTex1 fibrils has been reported, although a diversity of schematic or cartoon-style models has been published over the years. Here, we build on our above-presented polyQ amyloid core structures to construct an experiment-based molecular structure of Q44-HTTex1 fibrils. The Q44-HTTex1 construct is used to model the disease-relevant HD protein in a variety of experimental studies[20,21,41,53,55], as HD patients commonly have CAG repeat expansions that yield HTT proteins with approximately forty residues in their polyQ domain[2]. In experimental studies of Q44-HTTex1 fibrils formed in vitro, similar to our analysis of the polyQ_{15} peptide fibrils above, a large majority of the polyQ residues are observed to be buried in the fibril core, in a multi-nm-size block-like core architecture[41]. The polyQ-expanded HTTex1 protein fibrils were observed to form protofilament architectures[21,41] where the polyQ segment adopted a β-hairpin structure (Supplementary Fig. 1C)[8]. NMR on Q44-HTTex1 fibrils revealed a β-hairpin with a single turn and two ~20-residue-long β-strands (Supplementary Fig. 1C). This experimental finding was enabled by isotopic dilution studies of Q44-HTTex1 done via NMR, showing close intra-protein contacts between the "a"- and "b"-type strand backbones[8]. This

finding recapitulated an earlier 2D-IR study that reported β-hairpins in longer polyQ peptide aggregates lacking HTT flanking segments[30]. The presence of β-hairpins in (long) polyQ fibrils also was suggested by other mechanistic and mutational studies, in vitro and in cells[56–59]. Solid-state NMR studies with site-specific amino acid labels have indicated that the ordered β-sheet core extends from the final N17-domain residue (F17) to the penultimate glutamine in the polyQ segment[20,28]. Unlike the polyQ segment, the two polyQ-flanking segments of HTTex1 (see Fig. 1A) were found to be solvent exposed, lack β-structure, and display increased motion and disorder[19,21–23]. The short N-terminal N17 segment has been reported to adopt a random coil or α-helical structure, depending on context[28,60]. In fibrils, electron paramagnetic resonance (EPR) and ssNMR have shown the N17 segment to display partial order, and partly α-helical structure[19,21,28]. The longer C-terminal proline-rich domain (PRD) is disordered, except for the polyproline-II (PPII) helices at the locations of the two oligoproline segments.

Using these experimental data as input, we constructed the Q44-HTTex1 core base-architecture after the schematic model described earlier[8,20,21,41]. The 44-residue polyQ segment has a β-hairpin conformation with a tight β-turn, containing an even number of residues, and two long β-strands extending from F17 to the penultimate glutamine. A notable feature of polyQ protein aggregates is that the polyQ domains can come together in many different orientations and alignments, yielding a large propensity for heterogeneity and stochastic modes of assembly[20,23,41]. Here, a configuration was selected to construct a model that mimics our previous schematic model[41], with flanking domains evenly distributed on both sides of the fibril core. To this end, we attached the appropriate HTTex1 flanking domains to the terminal regions of the polyQ section, thereby achieving the formation of monomer building blocks as illustrated in the top panel of Fig. 6A. The thus constructed HTTex1 fibril was subjected to unrestrained all-atom MD simulations to assess its stability and monitor its structural dynamics. The fibril conformation obtained after a 5-μs simulation is shown in Fig. 6A.

## Structural analysis of HTTex1 fibril

The resulting structure (Fig. 6A) reveals interesting features that permit comparison to experimental studies. Firstly, consistent with the

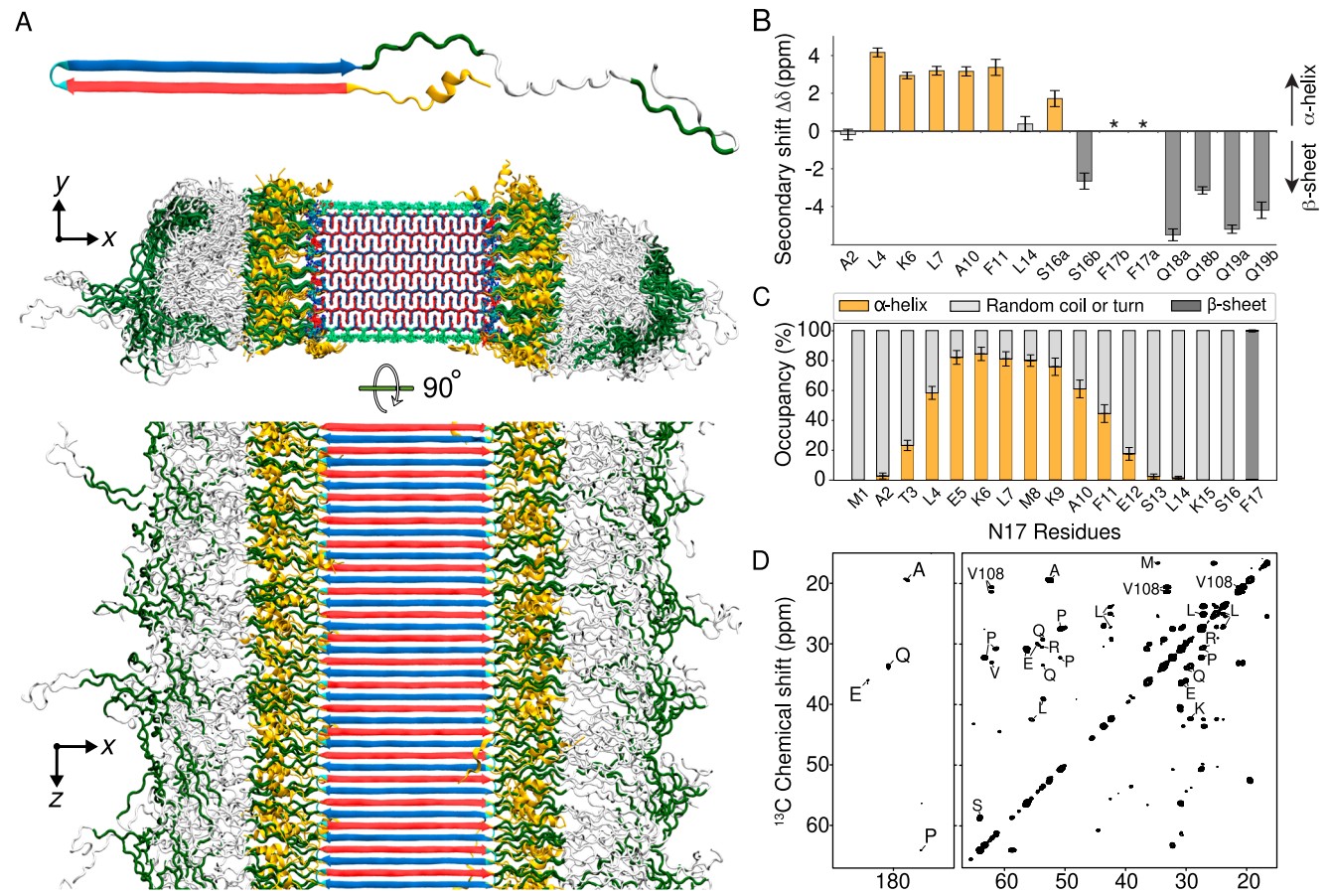

**Fig. 6 | Structure of Q44-HTTex1 amyloid fibril. A** Atomic-level structural model for mutant HTTex1 fibril. The top image shows a representative monomer within the fibril, with its β-hairpin polyQ segment to the left and the largely disordered flanking segments to the right. The middle image shows a cross-section, and the bottom image a top view of the fibril. The polyQ core is shown with the conformer-identifying red (for "a") and blue (for "b") β-strands, the N-terminal flanking segments yellow, and the C-terminal polyproline II helices dark green. Surface residues of the polyQ amyloid core are light green in the middle image. **B** Secondary chemical shift values for residues in the N-terminal end, indicating local α-helix (positive values) or β-sheet (negative values) conformations, replotted from previously reported work[21]. Doubled peaks indicating multiple co-existing conformations are marked with letters a and b. The asterisks mark F17, for which a peak ($^{13}C_\beta$) was not detected, but other resonances indicate β-sheet structure. Error bars reflect the s.d. in the chemical shift values[21]. **C** Secondary structure distributions observed for the 17 residues in the N-terminal flanking domain N17 during the last 500 ns of a 5-μs MD simulation (Amber14SB[86]). Error bars show s.d. across the occupancies (observed during the 500 ns) of the 140 individual HTTex1 monomers. The occupancies from the second-last 500 ns were identical within the error bars. **D** A 2D TOBSY ssNMR spectrum of Q44-HTTex1 fibrils, in which observed cross-peaks correspond to highly flexible residues outside the fibril core. Most, but not all, peaks originate from the C-terminal tail of the PRD domain. Spectrum was acquired at 600 MHz at 8.33 kHz MAS. Source Data files for panels (**B**–**D**) provided in ref. 85.

polyQ$_{15}$ fibril model, the HTTex1 polyQ core structure is found to be highly stable. A noteworthy observation lies in the β-turn conformation embedded within the β-hairpin structure: The simulation data underscores predominance of the type II turn over a type I' conformation (Supplementary Fig. 16), a finding that concurs with the experimental evidence gathered from ssNMR study of such compact turns[8]. As for polyQ$_{15}$ fibrils, a minority population of glutamines (the light-green side-chains of the middle image in Fig. 6A) is exposed to the solvent; these surface side-chains, as analyzed in Figs. 4 and 5 above, show a semi-rigid behavior. The constrained dynamics of these solvent-exposed glutamines stand in large contrast to the dynamic disorder of both non-polyQ flanking domains—whose disposition and structure are of substantial interest, given that they govern key structural and pathogenic properties of the protein and its aggregates[19–21,41,61,62]. Figure 6A illustrates the high level of disorder that appears in the MD ensemble of the flanking domains, manifested in both the N17 and the PRD. Although such pronounced disorder interferes with the detailed experimental study, its appearance in simulations fits with the dynamic disorder of this fuzzy coat observed in experiments, whether based on EPR, EM, or ssNMR[4,19–23].

The secondary-structure preferences of N17, polyQ, and PRD are summarized in Supplementary Fig. 17. There has been significant interest especially in the N17 segment, as it drives HTTex1 aggregation, but also harbors post-translational modifications that regulate HTTex1 (dis)aggregation and degradation[63–65]. The fate of N17 in the fibrils has remained somewhat opaque, with seemingly conflicting reports of the presence and absence of (partial) α-helicity. The obtained HTTex1 fibril model provides relevant molecular insights, as its N17 segment displays a mixed secondary structure content, with much disorder (Fig. 6C and Supplementary Figs. 18–21). Virtually all N17 residues are seen to show some propensity for disorder, such that a subset of the proteins in the modeled fibril has an N17 devoid of α-helical structure (Fig. 7D). Yet, in close to half of the protein monomers, an α-helix is observed within N17. These findings match well to ssNMR analysis of the structure of N17 in fibrillar samples:[21,28] Signals from an α-helical N17 were detected, but helicity was constrained to only part of the segment (Fig. 6B). The helical residues seen experimentally coincide remarkably well with the residues found to favor helicity in our model (Fig. 6C), providing further support for the validity of this structural ensemble. The observation that a significant part of N17 is not α-helical

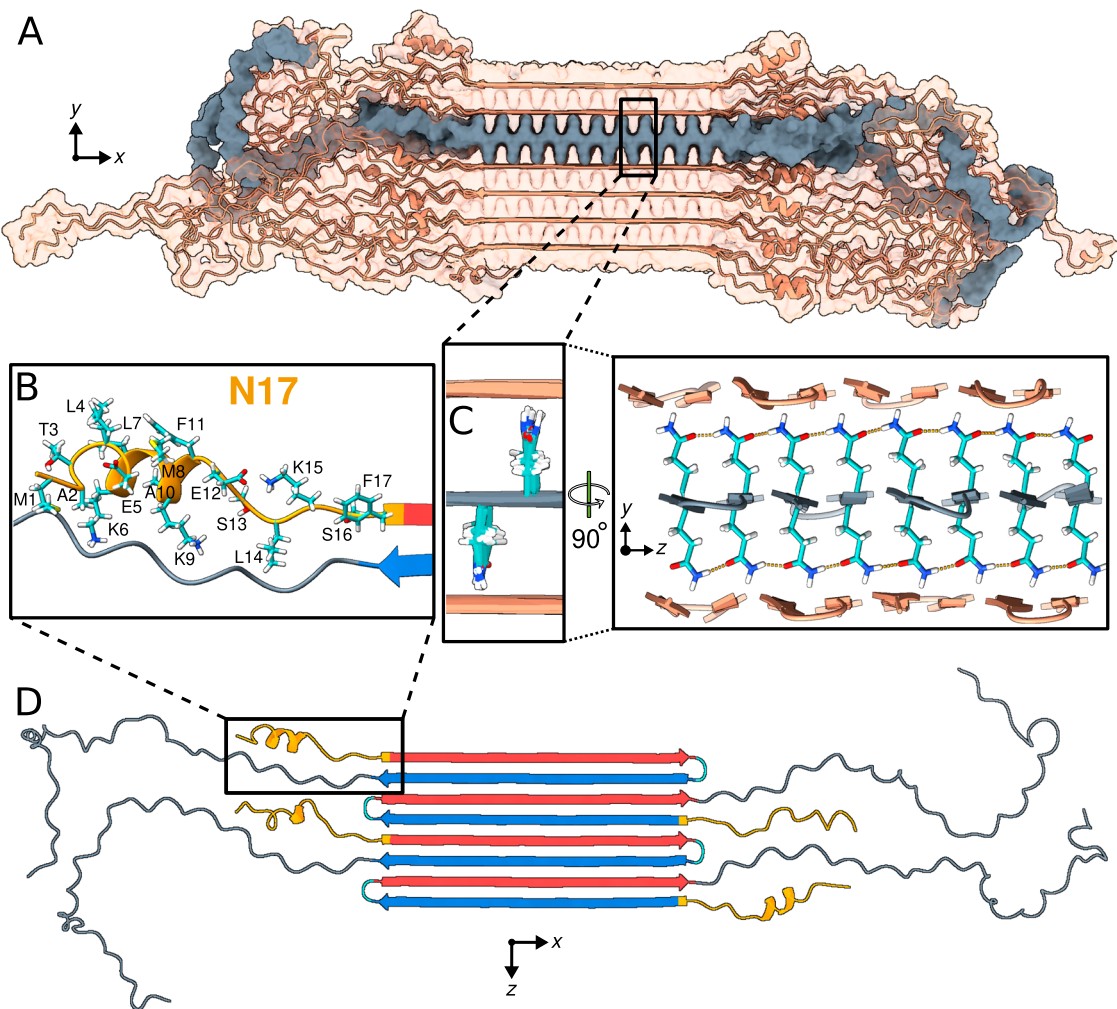

**Fig. 7 | Structural analysis of the Q44-HTTex1 amyloid fibril.** For an illustrative 3D exploration of Q44-HTTex1 fibril structure, see Supplementary Movie 1. **A** A graphical depiction (after 5-$\mu$s simulation in Amber14SB[86]) of the HTTex1 fibril. The region shaded in gray denotes a single sheet within the fibril's architecture. **B** An atomic view of the N17 domain within the fibril, naming the specific amino acids. **C** An atomic depiction of the glutamine side-chains within the fibril. The high stability of the fibril structure is primarily attributed to the extensive hydrogen bonding interactions among the glutamines, as depicted in the right panel. **D** Top view representation of the $\beta$-sheet highlighted in panel (**A**). A quartet of HTTex1 monomers is visible. The polyQ is color-coded for the type "a" (red) and "b" (blue) strands; the tight $\beta$-turn is cyan. The N17 and PRD domains are orange and gray, respectively. Note the structural variation between different monomers in the same fibril sheet, including in particular the range of helical content in the N17 domain.

finds experimental support as well[22,41,66]. Thus, also this experimental finding of N17 heterogeneity and plasticity is clearly recapitulated in the obtained fibril model: we observe an innate heterogeneity in structure and dynamics, even among protein monomers in the same fibril (Fig. 7).

C-terminal to the polyQ segment, the PRD is of biological interest given its role in reducing aggregation propensity, as well as its implication in toxic mechanisms. In HTTex1 fibrils made in vitro, the PRD is known to display a gradient of dynamics[19–22,41]. Experimentally, the PRD is relatively rigid proximal to the polyQ core, whilst its very tail end is highly flexible. The latter is evidenced by those residues showing up in INEPT-based MAS NMR measurements that are selective for highly flexible residues, such as the 2D INEPT-TOBSY spectrum on Q44-HTTex1 fibrils in Fig. 6D. These data are consistent with prior studies using similar methods as well as relaxation measurements[21,22,31]. The MD simulations indicate that the mobility of the PRD is not only constrained by its attachment to the rigid polyQ core, but also by pronounced PRD–PRD intermolecular interactions. Prior work has inferred a propensity for such interactions, especially in context of filament–filament interactions[21,32,41]. The current model suggest

that such interactions are also prominent in structuring the flanking domains of isolated protofilaments.

A notable feature of the HTTex1 fibril structural ensemble that was not a priori expected by these authors is that the flanking domains are not showing much interaction with the polyQ amyloid core surface. Throughout the simulation, the flanking domains display substantial disorder, but preferentially cluster together. This leaves the outer polyQ core surface easily accessible not only to solvent, but also to amyloid-binding molecules such as thioflavin-T (ThT) as well as PET ligands[10]. An in-depth and comprehensive visualisation of the described HTTex1 fibril structure can be found in the Supplementary Movie 1.

### Caveats and fibril polymorphism
It is important to note here that polyQ-based protein aggregates display a persistent and characteristic structural heterogeneity that is impossible to fully capture in practical MD simulations. Prior studies have discussed the propensity for polyQ segments to self-assemble in a disordered or stochastic fashion, due to the lack of sequence variation[8,20,67]. For instance, a polyQ chain extending an existing fibril can be added in different orientations. This variability is to some extent

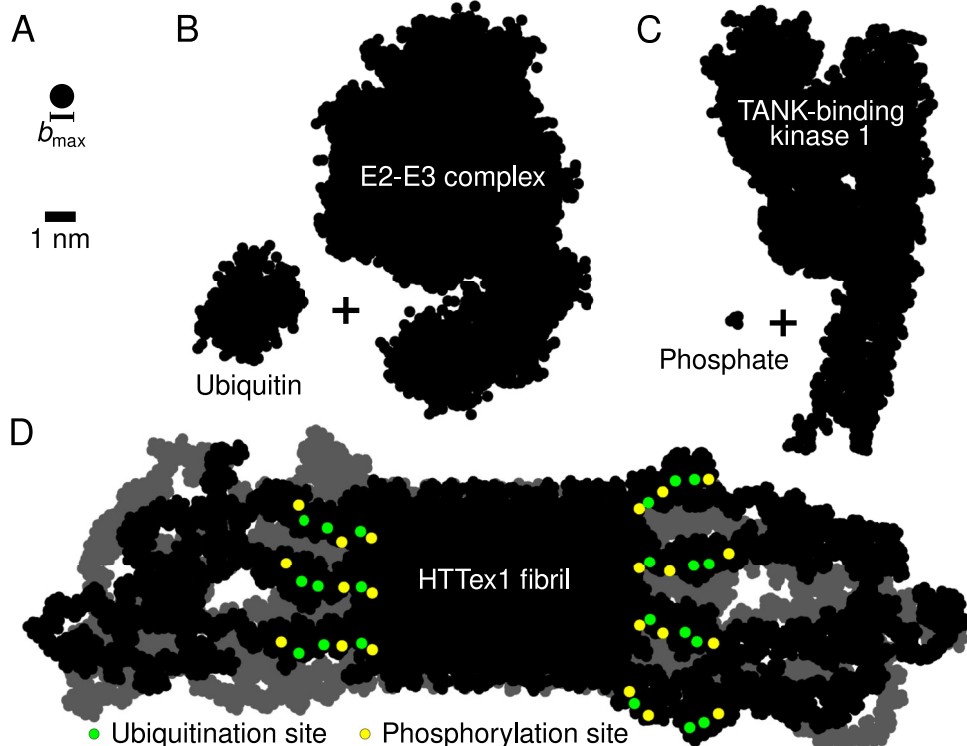

**Fig. 8 | PRD-domain brushes prevent post-translational-modification enzymes from reaching their target sites in HTTex1 fibrils. A** Size of the largest molecular species ($b_{max}$ = 0.8 nm) that can penetrate the polymer brush formed by the C-terminal PRD domains (as given by the polymer brush theory, see "Methods"). Silhouettes of the (**B**) E2-E3 Ubiquitin-Conjugating Complex (1C4Z) + ubiquitin (1UBQ), and (**C**) TANK-Binding Kinase 1 (6CQ0) + phosphate. **D** Silhouette of the HTTex1 fibril. The view is along the fibril growth direction. Shown are two consecutive monomer layers: the top layer in black, the layer behind it in gray. Post-translational modification sites in the top layer highlighted in color: the potential phosphorylation sites (residues T3, S13, and S16 in the N-terminal N17 domain) in yellow, and ubiquitination sites (residues K6, K9, and K15, also in the N17 domain) in green.

already displayed in our model: The alternating monomers in the single sheet in Fig. 7D have differently organized β-hairpins in their polyQ segment and their N17 segments on opposite sides of the filament. However, in real samples, one can expect a much more random patterning, which would vary at different locations among even a single fibril.

A similar stochastic feature that is explicitly missing from our model is that incoming polyQ segments can add to the fibril with register mis-alignments, resulting in shorter or longer β-strands in slightly different sequence positions, without a major energetic cost. This would be expected to yield fibril cores with varying fibril widths, local defects, and protein-to-protein structural variations[20,23,67] not accessible to more canonical amyloid fibrils formed by other proteins. This includes the recently reported cryo-EM structure of amyloid fibrils formed by the Orb2 protein (6VPS), a functional amyloid involved in memory formation (Supplementary Fig. 22)[49]. Although it is glutamine-rich in its amyloid core, the observed structure matches the typical in-register parallel fold and its glutamine torsion angles differ from those of the two polyQ conformers (Supplementary Fig. 22B–C). The presence of non-glutamine residues in the fibril core dictates a sequence-alignment that is absent in polyQ protein fibrils. In contrast, the latter fibrils are expected (and observed[23,41]) to display inherent structural variations between fibrils in a single sample, and even within single fibrils. Thus it appears that, like snowflakes, each HTTex1 fibril is unique (which prohibits the canonization of a single definitive protofilament structure) but still characterizable by well-defined structural features. This indistinguishability in local (atomic) structure of differing fibril architectures manifests for instance in the near-identity of ssNMR peak positions reported for polymorphic HTTex1 fibrils.

Indeed, like other amyloids, polyQ proteins form different fibril polymorphs depending on experimental conditions[21,32,68]. This structural variation imbues the fibrils with different degrees of cytotoxicity[32,68,69]. Notably, the HTTex1 polymorphs often reflect a type of 'supramolecular' polymorphism, with different supramolecular assemblies formed from similarly structured protofilaments[41]. The current model is expected to illuminate the atomic level conformation of one such protofilament, as it is based on experimental constraints that define its structure. Even inside a single protofilament, the unusual block-like architecture of polyQ fibrils permits variations in the number of sheets packed into a single protofilament[23]. Here, we also fixed this parameter, based on the dominant structures seen in one of our prior studies of Q44-HTTex1, but also this parameter certainly varies between and within samples.

Thus, in summary, in numerous different ways, the structure of HTTex1 is expected and observed to vary from sample to sample, from fibril to fibril, and even within single fibrils. This manifests in cryo-ET, cryo-EM, and AFM studies as fibrils that show variability in their structure, with much less order than many other amyloid fibrils. Capturing this diversity in a single set of MD simulations is impractical, but we consider the obtained structures as representative of the canonical or typical protofilaments seen in HTTex1 in vitro fibrils.

**Implications for HTT fibril interactions and properties**

A number of notable features of our model match or rationalize reported biological properties of HTTex1 fibrils. We obtained important insights into the surface-accessible molecular features of the HTTex1 fibrils' fuzzy coat. Fibril surface properties are crucial for their biological, possibly pathological, properties. In our fibril architecture, the N17 segment is found to reside outside the fibril core, where it

displays conformational and dynamic disorder. Crucially, however, N17 is tightly enclosed by the C-terminal PRD domains: Residues in N17 may be solvent-accessible, but are nonetheless largely inaccessible to larger macromolecules such as chaperones, kinases, ubiquitinases, and other potential N17 interaction partners, as can be unambiguously concluded from polymer brush theory (Fig. 8). Thus although in Fig. 7B the known phosphorylation sites T3, S13, and S16 may appear accessible, viewing this monomer in context (Figs. 7A, 8) underlines that the N17 segment is fully surrounded by the longer PRD segments. Similarly, the only HTTex1 ubiquitination sites involve lysines K6, K9, and K15 in N17, rendering them largely inaccessible in fibrillar HTTex1. The buried nature of N17 also rationalizes the low level of engagement by the TRiC chaperone:[34,70] Although known to bind this part of HTTex1, prior EM studies have shown TRiC to be unable to engage HTTex1 fibrils, except near the fibril ends. N17 is also implicated in membrane interactions, such that its preferential exposure at fibril ends may explain the latter to be engaged with the ER membranes[62,71,72].

Let us briefly discuss the implications of our models for HTTex1 fibrils formed by proteins with longer (and shorter) polyQ segments. Our HTTex1 fibril modeling focused on the HD-relevant Q44 that falls into the regime of common expansion lengths seen among patients. Yet, famously, mutant proteins can differ widely in their polyQ lengths. As illustrated in the polyQ$_{15}$ peptide fibril, and discussed elsewhere[38,66,69], proteins with shorter polyQ lengths can still form fibrils (at least in vitro). However, in such fibrils the polyQ segment may not form a $\beta$-hairpin, but instead occupy a single extended $\beta$-strand. Naturally, this would modulate the disposition of flanking segments on the fibril surface. Nonetheless, the qualitative architecture would remain unchanged. Conversely, HTTex1 with longer polyQ lengths, such as those associated with juvenile HD, would be expected to form amyloid cores featuring multiple turns, unlike the single-turn structures analyzed here for Q44-HTTex1.

Naturally—although we expect the obtained fibril structure to be a good representation of the fibrils present in our samples and also to have strong predictive and descriptive qualities for cellular HTTex1 aggregates—it cannot be excluded that cellular factors (ranging from chaperones to membrane interactions) may modulate the aggregation mechanism to the extent that the mature fibril architecture differs from the one obtained in vitro. Yet, unlike for other amyloid proteins, a remarkable feature of the HTTex1 ssNMR studies is that multiple groups have studied a variety of HTTex1 fibrils and always found the same signature spectra that are connected to the structural parameters used to construct our model[20–22,32,41]. Indeed, there is little evidence for a qualitative change in fibril architecture. Thus, we are inclined to expect that while cellular conditions may change certain details in the fibril structure, they would not fundamentally change the protofilament architecture.

The presented structural models of polyQ$_{15}$ and Q44-HTTex1 amyloid fibrils provide the best atomistic views of these disease-associated protein inclusions to date, derived from a multi-technique structural analysis through integrative modeling. The HTTex1 structure rationalizes a variety of experimental findings with notable biological and biomedical implications. The polyQ segment is mostly buried within the fibril, but the model reveals the structural and dynamical features of the minority of solvent-facing residues. These surface residues have proved challenging for experimental study, necessitating tailored and targeted approaches in, e.g., ssNMR[53]. A better structural understanding of this special polyQ surface will be useful in efforts to design polyQ-amyloid-specific binders, e.g., for PET imaging[10]. The visualization of the dynamically and structurally heterogeneous flanking domains enhance our understanding of their accessibility in the fibrils and pave the way for more in-depth analyses of their role in intracellular interactions with proteins and organelles.

## Methods

### Protein production and fibrillation

Mutant huntingtin exon 1 with a 44-residue polyQ core was expressed as part of a maltose binding protein (MBP) fusion protein, with the MBP attached to the N-terminus of HTTex1 to prevent aggregation[20,21]. The fusion protein MBP-Q44-HTTex1 was expressed in *Escherichia coli* BL21 (DE3) pLysS cells (Invitrogen, Grand Island, NY). Uniformly $^{13}C$ and $^{15}N$ labeled MBP-Q44-HTTex1 protein was expressed with $^{13}C$ D-glucose and $^{15}N$ ammonium chloride for MAS ssNMR studies. Then, cells were pelleted at 7 000 g, resuspended in phosphate buffered saline (PBS), pH 7.4 and lysed in presence of 1 mM phenylmethanesulfonyl fluoride (PMSF) by a HPL 6 (Maximator Benelux BV, The Netherlands). After that, cells were centrifuged at 125,000 × $g$ for 1 h using an Optima LE-80K ultra-centrifuge (Beckmann Coulter). The supernatant was filtered over Millex-GP syringe-driven 0.22 μm PES membranes (Millipore Sigma, Burlington, MD). The MBP-Q44-HTTex1 protein was purified by fast protein liquid chromatography (FPLC) using a 5 ml HisTrap HP nickel column (GE Healthcare, Uppsala, Sweden) with 0.5 M imidazole gradient (SKU I5513-100G, Sigma, St. Louis, MO) on an AKTA system (GE Healthcare, Chicago, IL). The imidazole was removed from the purified protein using an Amicon Ultra centrifugal filter with a regenerated cellulose membrane (Millipore Sigma, Burlington, MA). At least 3 washes with imidazole-free PBS buffer were done. Protein concentration was calculated from the absorbance at 280 nm. According to ProtParam tool by ExPasy[73] the extinction coefficient of the fusion protein is 66,350 M$^{-1}$cm$^{-1}$. Protein aggregation was initiated by addition of Factor Xa protease (SKU PR-V5581, Promega, Madison, WI) at 22 °C, in order to cleave off the MBP fusion tag[21,41] and release Q44-HTTex1. To prepare the ΔN15-Q44-HTTex1 samples, trypsin protease was used to sever the fusion protein[41], yielding fibrils in which most of the N17 segment is absent. After 3 days, the obtained mature fibrils were washed with PBS to remove the cleaved MBP.

### Samples for deuterium exchange MAS NMR studies

For the proton-deuterium-exchange (HDX) experiments, prior to cleavage, batches of the fusion protein were exchanged into either protonated or deuterated PBS buffer by solvent exchange (using Amicon centrifugal filters with 10 kDa cutoff). Next, the protein concentration was adjusted to 50 μM (with protonated or deuterated PBS, respectively). Factor Xa was added (1:400 molar ratio, protease:fusion protein), to permit three days of aggregation at 37 °C. Next, the protonated or deuterated fibrils were recovered and washed with matching PBS buffer. Just before the MAS NMR measurements, the fibrils were washed with PBS, either protonated or deuterated, to study the proton-deuterium-exchange process by MAS NMR.

### PolyQ peptide samples

For EM and NMR experiments, synthetic polyQ-based peptides were prepared and studied in their aggregated state. These peptides were obtained by solid-phase peptide synthesis (SPPS) from commercial sources, submitted to disaggregation protocols and permitted to aggregate in PBS buffer[8]. The aggregated polyQ15 peptide[51] studied by TEM had the sequence $D_2Q_{15}K_2$, synthesized by SPPS by WatsonBio (Houston, TX). The polyQ peptide used in the $^{15}N$-detected 2D NMR studies (Supplementary Fig. 7) had the sequence $K_2Q_{11}pGQ_{11}K_2$ (p indicates D-proline), with two sequential Gln residues outfitted with uniform $^{13}C$ and $^{15}N$ labeling, synthesized by the Yale University peptide facility[8]. The peptides were aggregated at 1 mg/mL concentration in PBS buffer (pH 7.4) at 37 °C. Aggregates were harvested by centrifugation after 2–3 weeks.

### Transmission electron microscopy (TEM)

Transmission electron microscopy (TEM) was performed on mature $D_2Q_{15}K_2$ peptide fibrils and Q44-HTTex1 fibrils. The fibrils were re-

suspended in MiliQ and then 5 $\mu$l of the sample was deposited on the plain carbon support film on 200 mesh copper grid (SKU FCF200-Cu-50, Electron Microscopy Sciences, Hatfield, PA). The grid was glow discharged for 0.5–1 min before adding the sample. After 30 s of the sample deposition, the excess MiliQ was removed by blotting, and immediately the negative staining agent 1% (w/v) uranyl acetate was applied. After 0.5–1 min, the excess stain was removed and the grid was air dried. The images were recorded on a Tecnai T12 or CM12 transmission electron microscope.

## NMR experiments

The hydrated U-$^{13}$C,$^{15}$N-labeled Q44-HTTex1 fibrils were packed into a 3.2 mm ssNMR rotor (Bruker Biospin) using a ultracentrifugal packing tool[74]. The fibrils were packed in the rotor by centrifugation at ≈130,000 × g in a Beckman Coulter Optima LE-80K ultracentrifuge equipped with an SW-32 Ti rotor for 1 h. Experiments were performed on a wide-bore Bruker Avance-I 600 MHz (14.1 T) spectrometer or Bruker Avance Neo 600 MHz (14.1 T) spectrometer, using triple-channel (HCN) 3.2 mm MAS EFree probes from Bruker. Data acquisition was done using Bruker Topspin software (version 4.1.3). Chemical shifts were indirectly referenced based on the $^{13}$C shifts of adamantane, and reported relative to aqueous DSS ($^{13}$C, $^{1}$H) or liquid ammonia ($^{15}$N). NMR spectra were processed, analyzed and plotted using NMRpipe (version 2019.217.13.13) and CcpNmr Analysis software (version 2.4)[75]. All experiments were acquired using two-pulse phase modulated (TPPM) proton decoupling of 83 kHz during acquisition[76]. The 2D $^{13}$C–$^{13}$C DARR experiments[77] on uniformly labeled Q44-HTTex1 fibrils (Fig. 1F) were performed using a 3-$\mu$s 90° pulse on $^{1}$H, 4-$\mu$s 90° pulses on $^{13}$C, a $^{1}$H–$^{13}$C CP contact time of 1 ms at 275 K, a DARR mixing time of 25 ms, and a recycle delay of 2.8 s. 2D NCA and NCO experiments were performed on U-$^{13}$C,$^{15}$N-labeled Q44-HTTex1 fibrils, as follows. 2D $^{13}$C–$^{15}$N NCO experiments (Fig. 1G) were done at 277 K using a 3-$\mu$s 90° pulse on $^{1}$H, 8-$\mu$s 180° pulse on $^{13}$C, $^{1}$H–$^{15}$N contact time of 1.5 ms, $^{15}$N–$^{13}$C contact time of 4 ms and recycle delay of 2.8 s. 2D $^{13}$C–$^{15}$N NCA experiments were done at 277 K using a recycle delay of 2.8 s, a 3-$\mu$s 90° pulse on $^{1}$H, 8-$\mu$s 180° pulse on $^{13}$C, 1.5 ms and 4 ms $^{1}$H–$^{15}$N and $^{15}$N–$^{13}$C contact times, respectively. In NCA and NCO experiments, the power levels for $^{15}$N and $^{13}$C during N–C transfer steps were 50 kHz and 62.5 kHz, respectively. The 2D refocused-INEPT $^{13}$C–$^{13}$C 2D spectrum of U-$^{13}$C,$^{15}$N-labeled Q44-HTTex1 fibrils (Fig. 6D) was obtained with total through-bond correlation spectroscopy (TOBSY; P9$_3^1$) recoupling, measured at MAS rate of 8.3 kHz, using a 6 ms of TOBSY mixing time, a 3-$\mu$s 90° pulse on $^{1}$H, 4-$\mu$s 90° on $^{13}$C, at a temperature of 275 K[78]. To observe and assign the backbone and side-chain protons of the fibril polyQ core, $^{1}$H–$^{15}$N heteronuclear correlation (HETCOR) experiments were performed on both polyQ model peptides (where only Gln were labeled) and fully $^{13}$C,$^{15}$N-labeled HTTex1 fibril samples. The former sample allows us to exclude contributions from non-Gln residues to the detected signals. These experiments were applied to aggregates of K$_2$Q$_{11}$pGQ$_{11}$K$_2$ peptides in which only two (sequential) Gln were labeled with $^{13}$C,$^{15}$N, which were previously shown to display the characteristic polyQ core signature[8]. To compare those to the polyQ core of HTTex1 fibrils, we employed aggregates formed from U$^{13}$C,$^{15}$N Q44-HTTex1 cleaved with trypsin (Supplementary Fig. 7). The $^{1}$H–$^{15}$N HETCOR experiments on both samples were done using a MAS rate of 13 kHz, 100 and 350 $\mu$s CP contact times, 4 s recycle delay, a 3-$\mu$s 90° $^{1}$H pulse, and at 275 K temperature. 100 kHz homonuclear FSLG $^{1}$H decoupling was applied during the t$_1$ evolution time. For the peptide fibrils, 128 scans (per t$_1$ point) were acquired; for the protein fibrils 64 scans. The HDX MAS NMR experiments were performed using a 700 MHz Bruker NMR spectrometer, equipped with a 1.3 mm fast-MAS HCN probe. 2D $^{15}$N–$^{1}$H spectra were obtained using 2 ms $^{1}$H–$^{15}$N and 1 ms $^{15}$N–$^{1}$H CP transfers and a recycle delay of 1.1 s. Relaxation measurements were performed at 60 kHz MAS using relaxation delays of 0, 10, 30, 50, 100, and 200 ms for $^{1}$H–$^{15}$N $T_{1\rho}$ measurements using a $^{1}$H–$^{15}$N

spin lock amplitude of 18 kHz, and relaxation delays of 0, 0.1, 1, 2, 4, 8, and 16 s for $^{1}$H–$^{15}$N $T_1$ measurements. The $T_1/T_{1\rho}$ trajectories were fit to single exponentials.

## MD simulations

Let us first describe general simulation details. All MD simulations were carried out on the fast, free, and flexible Gromacs engine[79]. All systems were first energy-minimized using steepest descent with one conjugate gradient step every 100 steps, then equilibrated through a 200-ps MD simulation in the NVT ensemble with positional restraints (1 000 kJ/mol/nm$^2$) on heavy atoms, followed by three 100-ps NPT runs with positional restraints (1000, 500, and 100 kJ/mol/nm$^2$) on heavy atoms. (To create independent replicates of the 30 polyQ amyloid core lattices, see Supplementary Fig. 4, we alternatively equilibrated them using dihedral restraints on the $\chi_1$ and $\chi_3$ angles: First through a 100-ps NVT run with 1 000 kJ/mol/rad$^2$, followed by four 100-ps NPT runs with 1 000, 500, 250, and 100 kJ/mol/rad$^2$.) The production MD simulations were done in the NPT ensemble, obtained through the Bussi–Donadio–Parrinello[80] thermostat (T = 300 K, $\tau_T$ = 0.2 ps) and the isotropic (for the polyQ lattice systems) or semi-isotropic (Q$_{15}$ and HTTex1 systems, $xy$ and $z$ coupled separately) Parrinello–Rahman[81] (P = 1 bar, $\tau_P$ = 2 ps, $\kappa_P$ = 4.5 × 10$^{-5}$ bar$^{-1}$) barostat. The van der Waals interactions were switched off between 1.0 and 1.2 nm; long-range electrostatics were treated via Particle Mesh Ewald[82,83] with fourth-order interpolation, a real-space cut-off at 1.2 nm, and size-optimized fast Fourier transform parameters (grid spacing of roughly 0.16 nm). Covalent bonds involving hydrogens were constrained to their equilibrium lengths by (fourth-order double-iteration) parallel linear constraint solver (P-LINCS)[84]. Timestep was 2 fs, Verlet neighbor lists updated every 20 fs with the neighbor list radius automatically determined. Input files containing the complete sets of simulation parameters used are permanently openly available on Zenodo[85].

Let us then describe the specific details for polyQ amyloid core lattices. We first describe building these systems. To build atomistic-resolution models of the internal structure of the polyQ amyloid core, we assumed it to comprise a lattice of antiparallel $\beta$-sheets stacked such that the Gln side-chains interdigitate. The pairs of side-chain dihedral angles ($\chi_1$, $\chi_3$) were set to satisfy the known characteristic of the interdigitating polyQ amyloid core:[8,26] the existence of side-chain–side-chain hydrogen bond interactions. We took the fibril-axis direction to align with the Cartesian coordinate $z$, and to be perpendicular to the $\beta$-strands (aligned with $x$). To construct the 3D lattice, we considered a minimal unit cell consisting of eight Gln residues arranged in a 2 × 2 × 2 pattern (Fig. 2A): Along $x$, the unit cell contains a minimal peptide chain segment of two amino acids (2 × 2 × 2), representing the alternating (odd/even, i.e., pointing above and below the $\beta$-sheet plane) residues of the $\beta$-strand. Along $z$, to describe an antiparallel $\beta$-sheet, minimum two $\beta$-strands are needed (2 × 2 × 2); in contrast to parallel in-register sheet structures that could be represented with a single repeating $\beta$-strand. Along $y$, the unit cell contains two distinct neighboring $\beta$-sheets to permit and probe distinct sheet–sheet interfaces (2 × 2 × 2). Figure 2A illustrates how each eight-Gln unit cell contains four Gln–Gln pairs, which establish backbone hydrogen bonds (shown in purple) as well as side-chain hydrogen bonds (in orange). All the hydrogen bonds are aligned roughly along the fibril axis, $z$. For each Gln–Gln pair, there are 8 distinct classes of ($\chi_1$, $\chi_3$) orientations that permit side-chain–side-chain hydrogen bond chains along $z$ (see Supplementary Fig. 2). As there are 4 Gln–Gln pairs in the unit cell, there are 8$^4$ = 4 096 possible plausible atomistic structures of the 8-Gln unit cell. Accounting for rotational and translational symmetry of the 3D lattice reveals, however, that at most 1 280 of these structures are unique. An important further consideration is that ssNMR experiments conclusively demonstrate that consecutive residues within each $\beta$-strand must adopt the same backbone conformations, as indicated by identical chemical shifts. As a result, each

distinct strand exclusively contains either type "a" or type "b" Gln residues (Fig. 1H). After applying this filter, the number of possible distinct unit cells is reduced to 30. For each of these 30 unit cells candidates, we performed all-atom MD simulations. The fully periodic MD simulation box was filled by 40 identical unit cells, that is, a total of $8 \times 40 = 320$ Gln residues, organized as a stack (along $y$) of 4 anti-parallel $\beta$-sheets, with each sheet comprising 8 $\beta$-strands of periodic (along $x$) $Q_{10}$ peptides, see also Supplementary Table 1.

Let us then describe the MD simulation details specific to polyQ amyloid core lattice simulations. Acknowledging the limitations of a classical mechanics approximation (force field) of an inherently quantum system, we employed in parallel three state-of-the-art MD force fields (one from each of the main force field families): AMBER14SB[86], OPLSAA/M[87], and CHARMM36m[88]. Gromacs version 2018.3 was used. During the production runs (10 $\mu s$ for position-restraint-initialized Amber14SB (Fig. 2B), 1 $\mu s$ for other force fields and/or dihedral-restraint initialization (Supplementary Fig. 4)), the stability of the 30 structural candidates was evaluated at 5 ns, 100 ns, 200 ns, and 1 $\mu s$. At each of these time points, only the candidates that maintained their stability above 0.9 (see Eq. (1)) were further continued. The stability $S(t)$ of the given structural candidate at simulation time $t$ was defined based on the $\chi_1(t)$ and $\chi_3(t)$ dihedral angles compared to the initial energy-minimized MD structure:

$$S(t) = \frac{N_1(t) + N_3(t)}{2N}, \tag{1}$$

where $N = 320$ is the total number of residues, and $N_i(t)$, $i = \{1, 3\}$ is the number of residues whose $\chi_i(t)$ is within 90° of its initial value at time $t$. For the stable candidates $M1$ and $M2$, the complete production trajectories (with frames saved every 2 ps) were used for analysis.

Let us then describe the MD simulation details of the solvated polyQ$_{15}$ fibril systems. $Q_{15}$ peptide fibrils were constructed from each of the two stable ssNMR-verified core models, $M1$ and $M2$. Two successive aspartic acid (DD) residues were added to the N-terminus and two lysines (KK) to the C-terminus of the $Q_{15}$ peptide to mimic the $D_2Q_{15}K_2$ peptide widely studied by experiments[8,24,27,50]. The N-terminus was further capped with an acetyl (Ace) group, and the C-terminus was set uncharged (with –COOH capping) to match the peptides used in our experiments. A 7-sheet fibril structure was constructed, with each sheet comprising eight Ace-$D_2Q_{15}K_2$ peptides. The simulation box was set up to form a quasi-infinite fibril (along $z$) under periodic boundary conditions and solvated with ~9 800 water molecules in a cuboid box of ~11 $\times 11 \times 3.8$ nm$^3$, see also Supplementary Table 1. State-of-the-art MD force fields AMBER14SB[86], OPLSAA/M[87], and the TIP3P[89] water model were used for calculations carried out on the Gromacs version 2021.3. Production run length was 1 $\mu s$; the complete production trajectories (with frames saved every 2 ps) were used for analysis.

Let us then describe the MD simulation details of the solvated Q44-HTTex1 fibril. Let us start with creating the system. The ssNMR data suggest that the HTTex1 aggregates display the same spectral patterns observed in polypeptide polyQ fibrils[8,38]. This indicates the presence of a common atomic polyQ core structure among them. Consequently, we constructed an atomistic-resolution structure of the mutant HTTex1 fibril utilizing the polyQ core model described in the preceding section. In contrast to fibrils composed of short polyQ segments, the Q44-HTTex1 fibrils exhibited a distinct structural characteristic, consisting of a $\beta$-hairpin structure with a single turn. Hence, the constructed core domain of our Q44-HTTex1 encompassed a stack of seven antiparallel $\beta$-sheets, each sheet comprising twenty 44-residue polyQ hairpins (Q44). The hairpin arms consisted of "a" and "b" $\beta$-strands connected by a two-residue $\beta$-turn modeled based on a type-I' tight turn, known for its strong preference towards adopting a $\beta$-hairpin structure (1KH0)[90]. The N-terminus of HTTex1, comprising 17 residues, was modeled utilizing the crystal structure of a single HTT(1–17) peptide in complex with the C4 single-chain Fv antibody (4RAV)[60]. The residue F17 was included within the $\beta$-sheet core, pairing with the penultimate glutamine residue, as supported by the findings of ssNMR investigations[28]. The PRD domain of the HTTex1, including 50 residues (P62–P111), was modeled as an end-to-end-distance-maximized random coil, with the two oligoproline (P62–P72 and P90–P99) segments in a polyproline-II-helix conformation. The terminal cappings were set to $NH_3^+$ and $COO^-$ for the N and C termini, respectively. The turns of neighboring Q44 hairpins were positioned on opposite sides of the fibril (Fig. 6A). The system was solvated with ~840 422 water molecules, resulting in a total of ~2,800,000 atoms, in a cuboid box of ~38 $\times 38 \times 19$ nm$^3$, see also Supplementary Table 1. The production MD simulations were then conducted for a duration of 5 $\mu s$ (with frames saved every 20 ps) on Gromacs version 2021.4 with AMBER14SB[86] protein force field and TIP3P[89] water.

The calculation of dihedral angles, distances, and protein secondary structures were done using the `gmx_angle`, `gmx_distance` (with `gmx_traj` used to obtain centers of mass for distances between groups of atoms), and `gmx_do_dssp` tools of GROMACS versions 2018.3 and 2021.4. Images of molecular structures were created using VMD 1.9.3;[91] except for Fig. 7 and the Supplementary Movie 1, which were created using ChimeraX 1.7[92].

### Polymer brush theory
The largest particle that a polymer brush in an ideal solvent can accommodate has the size[93,94]

$$b_{max} = \sqrt[4]{\frac{Na^2}{2\pi^3\sigma}}, \tag{2}$$

where $N$ is the number of monomer units per polymer, $a$ the monomer size, and $\sigma$ the grafting density. For a Q44-HTTex1 amyloid fibril, $N \approx 50$ and $a \approx 0.4$ nm are the number of residues in the PRD domain and the persistence length of an intrinsically disordered protein, respectively;[95] and $\sigma \approx 0.7$ nm$^{-2}$ is the density of PRD domains on the fibril sides where the polyQ-flanking segments reside. Using these values results in $b_{max} \approx 0.7$ nm; allowing for a good[95] instead of an ideal solvent gives the $b_{max} \approx 0.8$ nm used in Fig. 8.

### Reporting summary
Further information on research design is available in the Nature Portfolio Reporting Summary linked to this article.

## Data availability
Experimental and simulation data generated in this study have been deposited on the associated Zenodo repository[85] 13926360 that also contains the simulation input files and the initial and final coordinate files, as well as the Source Data files for Figs. 1F; 2B,D,G; 3C; 4B,C; 5; and 6B–D. Relevant chemical shifts are in the BMRB 27045 and BMRB 25146. In this work the following existing protein structures were used: 4RAV, 1C4Z, 1UBQ, 6CQ0, 6VPS, and 1KH0.

## Code availability
The associated permanently openly available Zenodo repository[85] 13926360 contains the in-house Python scripts used for MD analysis.

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

## Acknowledgements

This study was supported by funds from the CHDI foundation contract A-17778 (P.v.d.W., M.M.), the CampagneTeam Huntington (P.v.d.W.) and prior funding via NIGMS R01 GM112678 (P.v.d.W.). Part of the work has been performed under the Project HPC-EUROPA3 (INFRAIA-2016-1-730897; M.B.H.), with the support of the EC Research Innovation Action under the H2020 Program; in particular, M.B.H. gratefully acknowledges the support of Zernike Institute for Advanced Materials of the University of Groningen and the computer resources and technical support provided by SURF HPC. Majority of the calculations presented here were carried out on the MPG supercomputers RAVEN and COBRA hosted at MPCDF. Financial support by the Volkswagen Foundation (86110; M.S.M.) and by the Trond Mohn Foundation (BFS2017TMT01; M.S.M.) is gratefully acknowledged. This work benefited from access to the uNMR-NL facility at Utrecht University, an Instruct-ERIC center, with financial support provided by Instruct-ERIC (PID 18439; P.v.d.W.). We thank Dr. Alessia Lasorsa for recording some of the ssNMR experiments. M.B.H. thanks Dr. Matthias Elgeti for his support during the writing of this manuscript and for his valuable feedback.

## Author contributions

M.B.H. and J.O.D. performed and analysed MD simulations; M.B.H., J.O.D., and M.S.M. planned and designed the simulations; I.M., R.K, and G.J. performed and analysed NMR experiments; I.M. and G.J. prepared fibril samples; I.M., R.K, G.J., M.W., and P.C.A.v.d.W. designed experiments; M.B.H., I.M., P.C.A.v.d.W., and M.S.M. wrote the paper.

All authors reviewed and edited the manuscript and approved its final form.

## Funding

## Competing interests

The authors declare no competing interests.
