## [Transparent Peer Review file · Nature Communications]

Integrative determination of atomic structure of mutant huntingtin exon 1 fibrils implicated in Huntington disease

Corresponding Author: Professor Markus Miettinen

Version 0:

Reviewer comments:

Reviewer #1

(Remarks to the Author)

This manuscript by Helabad et al. reports a molecular structural model for huntingtin exon 1 fibrils containing a 44-Gln repeat (Q44-HttEx1), developed by applying molecular dynamics (MD) simulations to candidate models that were based primarily on structural constraints from previous publications. The structure of HttEx1 fibrils is a subject of considerable interest. Publication in Nature Communications may be appropriate, but the authors should address the following points:

1. The authors need to explain more clearly which pieces of experimental data are new and which have already been published. It appears that some of the ssNMR data in Figs. 1 and 2 may be new, but it also seems possible that these are new plots of older data. For any new experimental data, the authors should explain clearly how the new data provide new information about HttEx1 fibril structure.
2. What is the basis for assigning 1H-15N crosspeaks in Fig. 2F to "core" and "surface" Gln residues? Does Fig. 2F represent experimental support for the authors' structural model? If so, how?
3. Were all of the ssNMR measurements described in the Methods section actually included in this manuscript? The Methods seems to describe some measurements that may have been included in previous publications from the same research group but are not in the current manuscript.
4. What is the definition of "stability", plotted as S(t) in Fig. 2B?
5. The authors say 'As already noted, the "a" and "b" ssNMR signals reflect different χ_1 and ψ values in the two Gln conformers, but a similar χ_2 value. This is beautifully reproduced by the simulations of both polyQ15 models (Fig. 3C).' The phrase "beautifully reproduced" is not appropriate for a research paper. Moreover, since the ssNMR data allow broad ranges of sidechain torsion angles, it would be more accurate to say "the simulations are consistent with the data".
6. Why were positional restraints applied to heavy atoms in the early stages of MD simulations? Why not simply energy-minimize, then run unrestrained MD simulations? Were any restraints included on any atoms or any torsion angles or any interatomic distances during the "production" simulations? These questions should be answered explicitly in the text.
7. Most importantly, the authors' structural model assumes several major constraints on the HttEx1 fibril structure, namely that Gln residues within each beta-strand must have either the "a" or the "b" conformation (i.e. "a" and "b" conformations do not occur together in the same beta-strand), that different beta-strand segments from the same HttEx1 molecule lie in the same beta-sheet, and that the beta-strands are connected by beta-turns. The authors believe that these constraints are amply supported by experimental data in previous papers. My own assessment is that experimental data in previous papers is not entirely conclusive. Prior ssNMR, EPR, and infrared spectroscopy measurements do support antiparallel beta-sheets in polyQ and HttEx1 fibrils. However, support for the other constraints imposed by the authors does not seem very strong. For example, in the paper by Hoop et al. (PNAS 2016), crosspeaks between "a" and "b" Gln signals in 2D ssNMR spectra were weak or not detected when short 13C-13C mixing times were used. Such crosspeaks were strong when longer mixing times were used and were still observed when 13C-labeled molecules were diluted in 15N-labeled molecules. Hoop et al.

interpreted the absence of a-to-b crosspeaks with short mixing times as evidence against the existence of two different Gln conformations within a single beta-strand. They then interpreted the presence of a-to-b crosspeaks with long mixing times (even with isotopic dilution) as evidence for beta-hairpins. However, inter-residue crosspeaks are generally absent or weak when short mixing times are used, so the absence of strong a-to-b crosspeaks with short mixing times is not informative by itself. The a-to-b crosspeaks observed with long mixing times could then be either intra-strand or inter-strand, and they could be either intra-molecular or inter-molecular. When isotopic dilution is included, intra-strand a-to-b crosspeaks would be expected to remain. Inter-molecular contributions to the a-to-b crosspeaks would be attenuated, which would explain any reduction in the crosspeak intensities produced by isotopic dilution.

The recent paper by Nazarov et al. (JACS 2022) claims to provide evidence for beta-hairpins and beta-turns, based on features in some 2D class images from cryoEM studies of HttEx1 fibrils. However, the features that are described as beta-hairpins have the same intensities as background noise in the 2D class images. Therefore, these cryoEM studies do not represent strong evidence for beta-hairpin structures.

To address this major criticism, the authors need to explain in much greater detail what the experimental bases are for the constraints that they include in their structural modeling efforts. If they disagree with my assessment of the experimental evidence, they should explain why my assessment is necessarily incorrect. If alternative interpretations of various experimental observations are possible, the authors should mention the alternatives. A thorough explanation of the experimental bases for the authors' structural constraints will also make this manuscript more valuable to other research groups that are studying polyQ and HttEx1 fibrils.

Reviewer #2

(Remarks to the Author)

Helabad et al. used molecular modeling blended with data from fiber X-ray diffraction, fiber dimensions from EM, and structural constraints from ssNMR to provide structural models for mutant huntingtin exon 1 fibrils. They first studied the model peptide D2Q15K2 and then, using the knowledge from this model peptide, the biologically relevant Q44-HttEx1. Most efforts are spent on the arrangement of the sidechains in the fibrils for which ssNMR data was produced in the current study.

Beyond the orientation of the sidechains, not much new information with regard to the fibril structure is produced in the current study, as many structural details (like nontwisted filaments with antiparallel beta-sheets and beta-hairpins formed by the individual proteins) were previously uncovered by IR spectroscopy (10.1073/pnas.140158711) and cryo-EM (10.1021/jacs.2c00509).

The simulation study was performed very carefully and the results are overall nicely presented. However, it is not always clear which of the results were previously obtained by others and which were obtained in the current study, and which of the results are simulation results and which are from experiment.

The simulation results are presented very detailed, yet at various places too detailed in this reviewer's opinion. At the same time, relevant simulation details are not provided. For instance, only in the Methods section at the end it is mentioned that the simulations were done with three different force fields. In the Supplementary Material one can find that with one of the three force fields (Charmm36m) none of the 30 tested geometries were stable, but no further analysis was presented. The 30 initial geometries (or their characteristic angles) are not even shown. Furthermore, the reader is not informed from which of the force fields the results are shown in the manuscript.

However, my criticisms of the presentation of the results are all minor and could easily be remedied. The main weakness is that no novelties of major (biological) relevance were uncovered. The overall structure of the fibrils had already been determined previously, and the current study mainly fills in the gaps in side-chain alignments.

Reviewer #3

(Remarks to the Author)

Integrative determination of the atomic structure of mutant huntingtin exon 1 fibrils from Huntington's disease

The paper provides a meticulous analysis and integration of existing experimental data to develop a structural model of HttEx1 fibrils. The authors used fibrils produced either from recombinant mutant HTTEx1 protein or from synthetic peptides as an experimental basis for model generation. The fibril model established here suggests structural information about the polyQ fibril core as well as the relatively flexible N- and C-terminal domains.

In its current form, the study is too preliminary for publication in Nature Communications. However, if the authors are able to address the points raised below, I am happy to review the manuscript again.

Major concerns:

Novelty: The scientific advance in relation to previously reported structural studies (e.g., Nazarov et al., 2022; Boatz et al., 2020 and Matlahov et al., 2022) is too small.

The authors generated two structural models (M1 and M2) for Q44-HttEx1 fibrils through integration of previously reported data and all-atom molecular dynamics (MD) simulations. However, they do not provide specific experimental validation of their model predictions. For instance, it would be important to show how the exchange of glutamines through non-glutamines in the polyQ tract of HttEx1 affects the atomic resolution perspectives. Such experiments would enable

verification/falsification of specific predictions.

The title of the study is somewhat overstated. It suggests that huntingtin exon 1 fibrils prepared from patient tissues have been assessed. The authors have generated a structural model for experimentally generated fibrils. This is clearly an important undertaking. However, huntingtin fibrils formed in vivo may structurally and morphologically be distinct from in vitro generated ones (see also comments below).

Issues regarding a possibly misleading outlook.

1. Abstract clarification:

I suggest modifying the abstract to explicitly state that the authors incorporated pre-existing EM and ssNMR data. It is important to ensure the language conveys the limited contribution of original experimental data, emphasizing the use of existing data to impose constraints on their model.

2. Section and Figure 1 clarification:

- I recommend moving Figure 1, partly or in total, to supplementary material as it summarizes findings from other authors and does not contain original results.
- I suggest removing Cryo-ET data (panels B and C) from Figure 1 due to its lack of contribution to the model and the potential to mislead regarding in vivo representation. Furthermore, the authors do not take any data from pre-existing cryo-ET experiments to build their structural model.

Issues regarding overall assumptions

1. Avoiding to imply in-cell application:

- I recommend removing the in-cell aggregate data from Figure 1 and refraining from making assumptions about in-cell application to align with recent evidence regarding other amyloid fibrils. There is increasing evidence that the structural features of fibrils formed in vitro by proteins related to neurodegenerative disorders have limited resemblance to the structure of fibrils found in human patients.
- For that reason, I recommend avoiding speculations like the one found in line 248, in which authors compare cellular aggregates with in vitro aggregates.

Issues regarding the MDS methodology

1. Explanation of forcefield choice:

- I suggest explaining the choice of forcefields and reasons behind the discrepancy observed with the CHARMM36m.

2. Replicate number explanation:

- Did the authors carry out replicates of their simulations? I recommend including a section explaining the rationale behind the chosen number of replicates or the lack of them.

3. Addition of a Limitations section:

- I suggest including a section explicitly stating the limitations of the investigations in this study.

4. Figure 2 Panel D addition:

- I recommend adding a figure to display a couple of “failed” models and discussing why they showed low stability compare to M1 and M2.

5. Solvation of polyQ core simulations:

- I propose clarifying whether the polyQ core simulations were performed solvated or in vacuo. In case they were performed in vacuo, please provide references for the methodology where it states that the selected forcefields are suitable for those type of simulations.

Experimental data clarifications

1. HETCOR experiment:

I suggest clarifying whether HETCOR experiments were performed using HttQ44Ex1 or the peptide D2Q15K2.

2. References:

Appropriate references for experimental results described in line 96 are missing.

3. Figures 3 and 5:

I suggest clarifying the origin of TEM images in Figure 3 and whether the data in Figures 5B and 5D are original or taken from other authors (as can be inferred by the text).

4. Line 241:

I recommend clarifying or removing the comparison between the model and phase transitions observed in the nuclear pore complex that appears in line 241.

Version 1:

Reviewer comments:

Reviewer #1

(Remarks to the Author)

The authors have made major revisions to address my criticisms of the original version of this manuscript. They have clarified and expanded important parts of the text, added new data, and modified the figures. I am fully satisfied with these revisions. I have no further requests for revisions.

Reviewer #2

(Remarks to the Author)

The authors carefully addressed all the reviewers' comments. As a result, the manuscript was significantly improved.

However, I am still not fully convinced of the novelty of the results presented that would justify publication in Nature Commun. My main concerns are:

Previous data/knowledge from the same group: in 2016, part of the same group of authors had already published the existence of the "a" and "b" conformers for the side chains in the paper by Hoops et al. They had also shown that "a" and "b" must be adjacent in a beta-sheet. I understand that further NMR experiments were performed here to corroborate these earlier results and provide data on the fibril-water surface (see my comment below).

"a" and "b" conformers: MD simulations were used to create many structural models for these conformations. For this purpose, many simulations were performed, which led to the conclusion that only two of them are stable (referred to as "M1" and "M2" – please changed "devoted" to "denoted" to l. 123 on p. 5). However, if you look at Fig. S2, you can see - without running MD simulations (I'm a simulation scientist, so I have nothing against MD!) - that most of these conformations are not stable because no H-bonds can be formed between the side chains in these conformations.

Why did the authors not bother to provide the chi1 and chi3 values in Fig. S2 so that the reader can easily relate the information in Fig. S2, Fig. S3 and the main manuscript?

Another interesting question, in my opinion, would have been why the alternation of "a" and "b" conformers in a beta-strand (left figure in Fig. 1H) is not possible. I understand that this information was obtained from NMR, but MD would have provided the opportunity to give an explanation.

It is nice that Amber14SB and OPLSAA/M match, but Charmm36m is also a very good force field, also for the side chains. So it is a mystery to me why the performance of Charmm36m is exceptionally poor here. The authors should have checked what is happening here.

By the way, it is customary not to show the equilibration results that contain constraints, as was done here. This would have avoided misunderstandings with reviewer 3.

Another side comment: polyQ prefers antiparallel beta-sheets - which is emphasized by the authors on p. 5 - as this is the more stable beta-sheet compared to parallel alignment. And since all residues are Q, they can adopt this structure, whereas in other amyloids this would lead to many energetic mismatches between amino acids; hence the parallel orientation is adopted.

Surface-water interactions: It is not surprising that the conformations of the side chains are less ordered, since the side chains form H-bonds with each other in the fibril core and compete with H-bonds between water and the side chains at the surface. Thus, it is not a surprising finding; nonetheless, it should be reported as done by the authors.

In summary, I stand by my earlier assessment that the work was carefully done and should be published, but I don't see the major breakthrough that would justify publication in Nature Commun. However, if the other reviewers see it differently, I am happy to be proved wrong. The strength of the paper is that it brings together information from previous studies, supplements this with additional NMR data and provides structural models based on MD data. The MD data was very carefully prepared, but the structural models could have been derived with less effort.

Reviewer #3

(Remarks to the Author)

Integrative determination of the atomic structure of mutant huntingtin exon 1 fibrils implicated in Huntington's disease Helabad et al.

The manuscript was significantly improved and is now ready for publication in Nature Communications.

I have, however, two minor points that should be considered by the authors to further improve this manuscript:

1) The authors use the abbreviation Httex1 in their manuscript. However, the term Htt is utilized to indicate mouse huntingtin. Please see the recently reported nomenclature for HD research (DOI: 10.3233/JHD-240044). I assume that human HTT

exon 1 fragments were exclusively analyzed in this study. Thus, the term HTTEx1 (or HTTex1) should be used instead of HttEx1 in this manuscript.

2) Figure 8: in D black and gray colors were used to indicate HTTex1 fibrils. I do not fully understand what the gray coloring is good for. May be for clarity reasons coloring should be changed in Fig.8.

Responses to Reviewer Comments

We thank all reviewers for their specific and insightful comments and suggestions. Below we reproduce the comments verbatim and provide a point-by-point response to each query, along with specific information about accompanying changes in the revised manuscript.

Reviewer #1:

Reviewer #1 (Remarks to the Author):

This manuscript by Helabad et al. reports a molecular structural model for huntingtin exon 1 fibrils containing a 44-Gln repeat (Q44-HttEx1), developed by applying molecular dynamics (MD) simulations to candidate models that were based primarily on structural constraints from previous publications. The structure of HttEx1 fibrils is a subject of considerable interest. Publication in Nature Communications may be appropriate, but the authors should address the following points:

We thank the reviewer for their appreciation of the considerable interest in this topic and in this structure. We address the specific points below.

1. The authors need to explain more clearly which pieces of experimental data are new and which have already been published. It appears that some of the ssNMR data in Figs. 1 and 2 may be new, but it also seems possible that these are new plots of older data. For any new experimental data, the authors should explain clearly how the new data provide new information about HttEx1 fibril structure.

Indeed the original manuscript included both previously reported (NMR) data (used in the structural analysis) and also new NMR measurements. All reproduced data had been accompanied with statements in figure captions indicating that those data had been adapted from prior published work, where appropriate. Nonetheless, we have tried to make the differentiation clearer in the revised paper, by adding more statements in the text indicating reuse of published data (where relevant). In addition, we have removed (also based on other reviewer comments) some previously published data, such as the cryo-ET images originally present in Figure 1B-C. Finally, as discussed below, we included significant new NMR data in the revised manuscript, in order to address questions about our data interpretation, raised in the reviewer comments below.

2. What is the basis for assigning 1H-15N crosspeaks in Fig. 2F to "core" and "surface" Gln residues? Does Fig. 2F represent experimental support for the authors' structural model? If so, how?

To address these queries we have made substantial changes to our manuscript:

We have added (not previously published) 2D NMR experiments that we used to assign the peaks in the mentioned ¹H-¹⁵N spectrum from Fig. 2G (previously Fig 2F). These new data are in the new Supplementary Figure S7, which shows NMR spectra for two different polyQ-containing fibril samples. The SI figure and its caption explain the source of the assignments of the side chain and backbone peaks.

In the original paper, the identification of surface residues was based on an indirect deduction, derived from an identification of the peaks from the core, such that other peaks should be due to the surface. For a more rigorous dissection of this point, we now added new H/D exchange experiments combined with ¹H-detected MAS NMR. The selective replacement of exchangeable protons with deuterium, either in the core or on the fibrils surface, allowed us by ¹H-detected MAS NMR to unequivocally distinguish residues in either location. These new data unambiguously assign the core and surface residues. We have added these new data to the revised manuscript as new Fig. 5, supported by newly added Supplementary Figure S15.

We also improved our explanation of the relevance of the ¹H shifts of the core glutamine side chains for evaluating our model. To do so, we discuss more extensively recent work on Gln (and Asn) ladders studied by MAS NMR (by Wiegand et al; ref 43), and how the side chain ¹H shifts inform on H-bonding distances. We note that the a and b conformers are seen to form equally long H-bonds in our model (See Supplementary Figure S8), and that this fits to the detection of identical ¹H shifts for the two NH₂ side chains of the two Gln conformers. (We juxtapose this with variations in the corresponding shifts seen e.g. in HET-s fibrils studied by Wiegand and co-workers). To bolster this explanation we also adopted a clearer nomenclature from that work, which we explain and visualize in several revised figures (Fig. 2F-G; Supplementary Figs S7, S8, S15)

Finally, enabled by the new data mentioned above, we provide a more detailed discussion of the analysis of the surface residues and how they appear more dynamic than we had previously realised (both in the MD and in the new NMR data). Thus, these findings offer multiple types of support for our structural model. We also slightly expanded discussion of how a better understanding of the surface residues is of biomedical importance (for designing PET ligands, for instance; a topic we are working on ourselves; more below).

3. Were all of the ssNMR measurements described in the Methods section actually included in this manuscript? The Methods seems to describe some measurements that may have been included in previous publications from the same research group but are not in the current manuscript.

We have doublechecked the methods and confirmed that all methods in the manuscript were for spectra that were shown in the manuscript, and not previously published. To clarify this, we have added explicit mentions in the methods section to the relevant figures, to avoid confusion.

Methods in the original manuscript:

- 2D DARR spectrum - shown in Figure 1 (not previously published)
- 2D NCA/NCO spectra - shown in Figure 1 (not previously published)
- 2D TOBSY spectrum - shown in Figure 6D (not previously published)
- 2D HETCOR spectra - shown in Figure 2 and SI (not previously published)

Some of these data may resemble data in our prior work (since polyQ/HttEx1 spectra are highly reproducible), but these are spectra that were not previously published.

As noted above and below, we have greatly expanded the Methods section in the revised manuscript to provide more experimental details and to account for the addition of new data that was not present in the original manuscript.

4. What is the definition of "stability", plotted as S(t) in Fig. 2B?

We have clarified the caption of Fig. 2:

Original: (B) Stabilities of the 30 experimentally-feasible polyQ core candidates (represented by color-coded lines) as a function of MD simulation time.

Revised: (B) Stabilities (see Eq. (1) in Methods) of the 30 experimentally-feasible polyQ core candidates (represented by color-coded lines) as a function of MD simulation time.

5. The authors say 'As already noted, the "a" and "b" ssNMR signals reflect different χ_1 and ψ values in the two Gln conformers, but a similar χ_2 value. This is beautifully reproduced by the simulations of both polyQ15 models (Fig. 3C).' The phrase "beautifully reproduced" is not appropriate for a research paper. Moreover, since the ssNMR data allow broad ranges of sidechain torsion angles, it would be more accurate to say "the simulations are consistent with the data".

We have made the change as suggested, while also more explicitly noting that the mentioned experimental data are from prior published work (as requested by the reviewers in other queries):

Original: As already noted, the "a" and "b" ssNMR signals reflect different χ_1 and ψ values in the two Gln conformers, but a similar χ_2 value. This is beautifully reproduced by the simulations of both polyQ₁₅ models (Fig. 3C).

Revised: Experiments⁸ have shown that the "a" and "b" ssNMR signals reflect different χ_1 and ψ values in the two Gln conformers, but a similar χ_2 value. The simulations of both polyQ₁₅ models are consistent with these NMR data (Fig. 3C).

6. Why were positional restraints applied to heavy atoms in the early stages of MD simulations? Why not simply energy-minimize, then run unrestrained MD simulations? Were any restraints included on any atoms or any torsion angles or any interatomic distances during the "production" simulations? These questions should be answered explicitly in the text.

The gentle initiating protocol (of initially applying positions restraints and then gradually removing them) was chosen to assure that no *de-facto*-inconsequential incompatibilities between an ideal candidate structure and the force field would lead the structure losing stability during the first few steps of MD simulations. This is akin to the standard protocol of simulations of folded proteins, where, after energy-minimizing the whole system, a short simulation is run with the protein heavy atoms positionally restrained such that the water molecules can truly equilibrate and so will not artificially disturb the folded protein structure.

The production simulations were completely unrestrained. So to answer the reviewer's question explicitly: No restraints were applied on any atoms, torsion angles, or interatomic distances.

We have clarified these points explicitly in the caption of Fig. 2B:

Original: The three blue-shaded sections indicate position restraints of 1 000, 500, and 100 kJ/mol over consecutive 100-ps periods.

Revised: The three blue-shaded sections indicate the gentle protocol chosen for initiating simulations from the energy-minimized ideal structures: Decreasing position restraints (of 1 000,

500, and 100 kJ/mol/nm²) over three consecutive 100-ps periods led to the unrestrained simulation.

7. Most importantly, the authors' structural model assumes several major constraints on the HttEx1 fibril structure, namely that Gln residues within each beta-strand must have either the "a" or the "b" conformation (i.e. "a" and "b" conformations do not occur together in the same beta-strand), that different beta-strand segments from the same HttEx1 molecule lie in the same beta-sheet, and that the beta-strands are connected by beta-turns. The authors believe that these constraints are amply supported by experimental data in previous papers. My own assessment is that experimental data in previous papers is not entirely conclusive. Prior ssNMR, EPR, and infrared spectroscopy measurements do support antiparallel beta-sheets in polyQ and HttEx1 fibrils. However, support for the other constraints imposed by the authors does not seem very strong.

For example, in the paper by Hoop et al. (PNAS 2016), crosspeaks between "a" and "b" Gln signals in 2D ssNMR spectra were weak or not detected when short ¹³C-¹³C mixing times were used. Such crosspeaks were strong when longer mixing times were used and were still observed when ¹³C-labeled molecules were diluted in ¹⁵N-labeled molecules. Hoop et al. interpreted the absence of a-to-b crosspeaks with short mixing times as evidence against the existence of two different Gln conformations within a single beta-strand.

The above comments from the reviewer suggest that our identification of the lack of a-to-b variation within each β-strand would stem from ¹³C-¹³C 2D ssNMR analyses of the fibrils. This reflects a misunderstanding: the cited paper indeed relies heavily on such ¹³C-¹³C spectra, but does so to detect the β-hairpin motif, not to establish the a-b sequential ordering within strands. Indeed, ¹³C-¹³C spectra are a poor method to detect such interactions, exactly as the reviewer notes. Instead, we (and others) have relied on backbone-mediated NCOX and NCACX spectra to reach this conclusion, which is a more suitable approach. These types of spectra were reported in our 2016 Hoop et al. paper (ref. 8), but also discussed (again) in our 2020 Boatz et al. paper in JMB (ref. 37; see SI of that paper). Notably, also the Siemer group has reported such data, and came to analogous conclusions, in their paper (Ref. 21: Isas et al. Biochemistry 54 3942 (2015) 10.1021/acs.biochem.5b00281; see e.g. Fig. 3 in their paper).

In the revised manuscript we tried to more clearly explain the origin of this conclusion, citing also the papers mentioned above. To aid with this, we have created a new Figure in the Supplementary Materials that summarizes some of the underlying prior data (Supplementary Figure S1). In the SI caption we briefly summarize the experimental explanation. In the main text we note the following, referring to that figure:

page 4: *“A key additional consideration is that ssNMR has unambiguously shown sequential residues within each β-strand to have the same backbone conformations (based on identical chemical shifts; Supplementary Figure S1B).^{8,21,35} Thus the “a” and “b” conformers strictly occupy distinct strands...”*

In the Figure S1 caption we also refer readers (and reviewers) to a recently published review article that provides a summary of the existing solid-state NMR studies of polyQ / HttEx1 amyloids (ref. 35; Van der Wel, P. C. Biochemical Society Transactions 52, 719–731 (2024).)

They then interpreted the presence of a-to-b crosspeaks with long mixing times (even with isotopic dilution) as evidence for beta-hairpins. However, inter-residue crosspeaks are generally absent or weak when short mixing times are used, so the absence of strong a-to-b crosspeaks with short mixing times is not informative by itself. The a-to-b crosspeaks observed with long mixing times could then be either intra-strand or inter-strand, and they could be either intra-molecular or inter-molecular. When isotopic

dilution is included, intra-strand a-to-b crosspeaks would be expected to remain. Inter-molecular contributions to the a-to-b crosspeaks would be attenuated, which would explain any reduction in the crosspeak intensities produced by isotopic dilution.

The identification of β -hairpins in our prior paper (ref. 8; Hoop et al) was based on quite careful, semi-quantitative analysis of spectra with different isotopic dilution levels. Our approach was enabled by matching data obtained on control samples, and the resulting analysis is in our view quite robust and reliable. We see no reason to doubt our conclusion that β -hairpins are present in the Q44-HttEx1 fibrils we studied. (NB. We have unpublished studies in which we continue those experiments to probe β -hairpins in other types of fibrils). Notably, a prior paper from a different group took a conceptually analogous approach, combining isotopic dilution with 2D IR spectroscopy to also detect β -hairpins in aggregates formed by polyQ model proteins. (Unlike our studies, those protein models lacked the flanking domains of the Htt protein implicated in HD). This paper from the group of Martin Zanni is cited and discussed in the paper (Buchanan et al; 10.1073/pnas.1401587111; ref. 29). We expanded the mention of that work in the revised paper (page 10). Finally, β -hairpin formation is more broadly implicated in polyQ aggregation, in mechanistic studies in vitro (e.g. several papers by Ronald Wetzel), but also in recent cellular studies by the group of Randall Halfmann in ELife (10.7554/eLife.86939.3). We have also cited a few of these studies, mentioning the indirect support from mechanistic and mutational studies for the presence of β -hairpins in polyQ fibrils. Given these considerations, we do not see a need to revise our conclusions regarding this point, although we did try to clarify our explanations and considerations in a few locations in the text. Most notably on page 10, we now write:

“This experimental finding was enabled by isotopic dilution studies of Q44-HttEx1 done via NMR, showing close intra-protein contacts between the a- and b-type strand backbones.⁸. This finding recapitulated an earlier 2D-IR study that reported β -hairpins in longer polyQ peptide aggregates lacking Htt flanking segments²⁹. The presence of β -hairpins in (long) polyQ fibrils also was suggested by other mechanistic and mutational studies, in vitro and in cells.^{58-61”}

The recent paper by Nazarov et al. (JACS 2022) claims to provide evidence for beta-hairpins and beta-turns, based on features in some 2D class images from cryoEM studies of HttEx1 fibrils. However, the features that are described as beta-hairpins have the same intensities as background noise in the 2D class images. Therefore, these cryoEM studies do not represent strong evidence for beta-hairpin structures.

Indeed, the cryo-EM data in the referenced paper by Nazarov et al. suffer from the inherent heterogeneity of polyQ protein fibrils, and the density for the turn region is indeed hard to distinguish from the noise. We do not disagree with this reviewer, and have tried to clarify in our revised manuscript that we see the cryo-EM results as supporting the general architecture of the polyQ core, without claiming support for actual β -hairpins (even though that is differently described in the original Nazarov paper).

On page 2, we now include the following edited sentence: *“Subsequently, a cryo-EM study of HttEx1 fibrils²² produced a medium-resolution density map, which—although it lacked the detail for a de-novo atomistic structure—was interpreted to be consistent with the abovementioned model architecture: the amyloid core has a block-like structure assembled from multiple layers of tightly packed β -sheets (as schematically shown in Fig. 1D–E).”*

To address this major criticism, the authors need to explain in much greater detail what the experimental bases are for the constraints that they include in their structural modeling efforts. If they disagree with my assessment of the experimental evidence, they should explain why my assessment is necessarily

incorrect. If alternative interpretations of various experimental observations are possible, the authors should mention the alternatives. A thorough explanation of the experimental bases for the authors' structural constraints will also make this manuscript more valuable to other research groups that are studying polyQ and HttEx1 fibrils.

In an attempt to address the concerns of this reviewer, we have expanded the explanation of the origins of used structural information throughout the paper. In addition, we added a new SI figure that summarizes key prior experimental data in more detail (Supplementary Fig. S1). We decided to put this in the SI, in accordance with the concerns from other reviewers that the manuscript was perceived to contain too many previously published data (although this was in part a misunderstanding). This figure summarizes the most important structural constraints obtained from prior solid-state NMR studies, although we do not have enough space to re-analyze each data point in detail. For this we refer (in the figure's caption) readers to the original work, but also to a new review article that we recently published and that precisely aims to provide a good overview of the (NMR-based) data available on polyQ/HttEx1 fibrils (Van der Wel 2024; DOI 10.1042/BST20230731; ref. 35). The new SI figure includes an illustration and explanation of the data supporting the β -hairpin motif, the long interrupted strand structures and other key parameters. We hope that our responses and clarifications satisfactorily address the concerns of reviewer 1 about the rigor of the underlying experimental data.

Reviewer #2:

Reviewer #2 (Remarks to the Author):

Helabad et al. used molecular modeling blended with data from fiber X-ray diffraction, fiber dimensions from EM, and structural constraints from ssNMR to provide structural models for mutant huntingtin exon 1 fibrils. They first studied the model peptide D2Q15K2 and then, using the knowledge from this model peptide, the biologically relevant Q44-HttEx1. Most efforts are spent on the arrangement of the sidechains in the fibrils for which ssNMR data was produced in the current study.

We see the side chain arrangement as a major aspect of the structure of the polyQ fibril core and consider it thus of significance and worth discussing in some detail. That said, based on the comments by this reviewer we have expanded discussion of other topics in the manuscript, such as an expansion of the role of the fibrils' fuzzy coat. This manifests for instance in the addition of a new main text figure (Figure 8) in which we visualize and analyze the relevance of the HttEx1 flanking domains for biological properties of the fibrils. (More responses to these and related comments from this reviewer are below)

Beyond the orientation of the sidechains, not much new information with regard to the fibril structure is produced in the current study, as many structural details (like nontwisted filaments with antiparallel beta-sheets and beta-hairpins formed by the individual proteins) were previously uncovered by IR spectroscopy (10.1073/pnas.140158711) and cryo-EM (10.1021/jacs.2c00509).

We respectfully disagree with the assessment that our work does not add new information beyond the mentioned works. The prior IR paper only examined the “structural motif” of polyQ fibrils (Buchanan et al. *Structural motif of polyglutamine amyloid fibrils discerned with mixed-isotope infrared spectroscopy* PNAS 2013; cited as ref. 29), and did not yield a robust atomic-resolution structure. Moreover, it did not study the kinds of HttEx1 fibrils relevant for Huntington's disease, but only analyzed polyQ model peptides/proteins that lacked the Htt flanking segments. The mentioned recent cryo-EM paper (Nazarov et al. *Structural Basis of Huntingtin Fibril Polymorphism Revealed by Cryogenic Electron Microscopy of Exon 1 HTT Fibrils* JACS 2022; ref. 22) did address HttEx1 fibrils, but focused on fibril polymorphism on the architectural level, without yielding an atomic structure model. Here we also point to Reviewer 1's comments (above) on the fact that the electron density map from that cryo-EM paper has a limited signal-to-noise, which prevented it from defining a de-novo or atomic-level structure.

Neither of the mentioned papers claims to have solved the atomic structure of the fibrils, even if they may include images of structural models. The basis of the structural visualizations in the Buchanan et al. 2D IR paper was obtained from the SI of an earlier publication (Schneider et al. JMB 2011), where it was constructed by simulated annealing in CNS to illustrate one possible hypothesis for the observed polyQ shift populations (which we note as conformers a and b). Notably, none of these prior illustrations were based on, or assessed against, any high-resolution structural constraints—such as the angle constraints used in our work. As a consequence, neither the original construction by Schneider et al., nor its MD-simulated version by Buchanan et al., fulfills our ssNMR-derived constraints (Fig. R1 for review only; prepared with coordinates kindly provided by respective authors of said papers).

[figure redacted]

Figure R1. Comparison of the backbone (φ , ψ) dihedral angles in three polyQ-core models. The scatter points show angles of the construction by Schneider et al. (in orange) and its MD-simulated version by Buchanan et al. (green); the panels on top (for φ) and right (for ψ) show the angle distributions in our model M1 for Gln types "a" (red) and "b" (blue). The dot-dashed lines show our ssNMR-derived constraints for the ψ dihedral angles of the Gln types "a" (red) and "b" (blue). These data are based on coordinate files kindly provided by the respective authors.

Methodologically, Schneider et al. obtained their model by simulated annealing under heavy constraints imposed (1) on dihedral angles to keep them β -sheet-like (but not derived from experimental measurements!), (2) on planarity of neighbouring β -strands, (3) on distances between backbone and sidechain amides and carbonyls to enforce H-bond lengths, and (4) on distances between C_{α} atoms to enforce a 8.3-Å stacking distance between sheets. Consequently, when Buchanan et al. removed these constraints, the structure underwent considerable changes, indicating a lack of long-term stability—behaviour not to be expected from the true structure of the polyQ core. Notably, our structures M1 and M2 do display such long-term stability (Fig. 2B).

As already noted, the 2D IR paper (Buchanan), though elegant and supportive of our β -hairpin structure, has another major difference with the current paper. It studied polyQ model peptides, without the full context of the HttEx1 protein fragments (implicated in HD). The ‘flanking domains’ outside the polyQ core play major roles in the aggregation process, are the sites of PTMs that modulate protein aggregation and toxicity, and are targeted by chaperones that prevent or even undo aggregation (e.g. DOI [10.1093/hmg/ddaa001](https://doi.org/10.1093/hmg/ddaa001) ; [10.1074/jbc.RA118.004621](https://doi.org/10.1074/jbc.RA118.004621) ; [10.1073/pnas.1320626110](https://doi.org/10.1073/pnas.1320626110) and many more). As such, it is essential to determine fibril structures for proteins that feature these segments. Notably, these flanking domains were present in the cryo-EM-studied protein fibrils, but formed a ‘fuzzy coat’ that proved too disordered to be resolved in detail by EM alone.

Finally, the reviewer seems to consider the ‘orientation of the sidechains’ unimportant. We strongly disagree—determining the structure of polyQ amyloid necessitates an understanding of the side chain conformations. As noted, prior models lacked any input constraints on, or tests

against, such details, and thus by necessity have produced arbitrarily configured polyQ models. Here, we for the first time were able to deploy experimental constraints to define the polyQ core structure in context of HD-related HttEx1 fibrils.

In summary:

- prior studies did not include proper experimental constraints needed to model polyQ amyloid core structure on the atomic level.
- when tested, previously reported ‘models’ are not consistent with currently available experimental data.
- prior studies did not reveal (or even include) the essential and important flanking segments of HttEx1, which are of key biological importance for HD biology.
- Atomic-level insight into HttEx1 fibrils is essential for understanding the protein misfolding process in HD, targeting aggregates with e.g. PET ligands (see below), or for analyzing and understanding (cellular) interactions of HttEx1 fibrils.

The simulation study was performed very carefully and the results are overall nicely presented. However, it is not always clear which of the results were previously obtained by others and which were obtained in the current study, and which of the results are simulation results and which are from experiment.

We wish to thank the reviewer for lauding our simulations and overall presentation. In light of the comments by this and other reviewers, we have revisited (and partially restructured) the manuscript to make a clearer distinction between new and old data. Here, we also refer to our responses to reviewer 1, above. We moved some older experimental data to a new SI figure (Supplementary Fig. S1) that makes an effort to explain the (prior) data underpinning our structural modelling efforts. We also added clarifying statements in many places, to more explicitly refer to re-used data as “previously published” (or similar phrases).

As noted above/below, we have also added multiple kinds of new experimental data in the revised text, in both main and SI figures.

We hope that our revisions sufficiently clarify the difference between new and old data, and between experimental and simulation results.

The simulation results are presented very detailed, yet at various places too detailed in this reviewer’s opinion. At the same time, relevant simulation details are not provided. For instance, only in the Methods section at the end it is mentioned that the simulations were done with three different force fields. In the Supplementary Material one can find that with one of the three force fields (Charmm36m) none of the 30 tested geometries were stable, but no further analysis was presented. The 30 initial geometries (or their characteristic angles) are not even shown. Furthermore, the reader is not informed from which of the force fields the results are shown in the manuscript.

Following Reviewer’s suggestion, we added in the Supplementary Material a table showing the characteristic torsion angles of the 30 initial geometries (Supplementary Fig. S3B), and refer to this figure on page 5 of the main text.

We have also clarified in the main text that three different force fields were tested. We do this by stating explicitly in the corresponding figure captions, for which force field the results are shown, for which other force fields corresponding simulations were run, and in which Supplementary Figure these results are shown; see below for the changes done in each caption.

In the caption of Fig. 2:

Revised: Panels B–E from the Amber14SB⁴⁴ force field; for OPLSAA/M⁴⁵ and CHARMM36m⁴⁶ see Supplementary Figures S4, S5.

In the caption of Fig. 3:

Original: The data were obtained from 1- μ s MD simulations.

Revised: The data were obtained from 1- μ s MD simulations (using the OPLSAA/M⁴⁵ force field; for Amber14SB⁴⁴ see Supplementary Fig. S10).

In the caption of Fig. 4:

Original: (B) Side-chain dihedral angle distributions for the buried Gln residues, and (C) for the Gln residues on the fibril surface.

Revised: (B) Side-chain dihedral angle distributions for the buried Gln residues and (C) for the Gln residues on the fibril surface (Amber14SB⁴⁴; for OPLSAA/M⁴⁵ see Supplementary Figures S13, S14).

In the caption of Fig. 6:

Original: (C) Secondary structure distribution of the 17 residues in the N-terminal flanking domain during the last 500 ns of 5- μ s MD simulation.

Revised: (C) Secondary structure distribution of the 17 residues in the N-terminal flanking domain during the last 500 ns of 5- μ s MD simulation (Amber14SB⁴⁴).

In the caption of Fig. 7:

Original: (A) A graphical depiction of the HttEx1 fibril.

Revised: (A) A graphical depiction (after 5- μ s simulation in Amber14SB⁴⁴) of the HttEx1 fibril.

In addition, we added further force-field-related analysis: new Supplementary Fig. S4 shows the stability-vs-time plots for all three force fields in two fully independent replicates (using initial position restraints and using initial dihedral restraints); and new Supplementary Fig. S5 shows the dihedral angle distributions for the stable models M1 and M2 in OPLSAA/M.

However, my criticisms of the presentation of the results are all minor and could easily be remedied. The main weakness is that no novelties of major (biological) relevance were uncovered. The overall structure of the fibrils had already been determined previously, and the current study mainly fills in the gaps in side-chain alignments.

We again reiterate our disagreement about this point. First, as discussed above, prior papers have not yielded a proper atomic model (based on actual experimental constraints) for polyQ amyloid, let alone for a disease-relevant HttEx1 fibril.

Second, the obtained model and information have great relevance to our biological understanding of important aspects of Huntington's disease and its molecular mechanisms. In the revised text we further clarify this point. We have added more discussion of the biological insight provided by our fibril structure. For instance we comment on the mechanistic understanding it brings on explaining the resilience of httEx1 fibrils against cellular detoxification mechanisms. To this end we have added a new schematic figure (Fig. 8), and modified text in the section *Implications for Htt fibril interactions and properties* :

Original: In our fibril architecture, the N17 segment is found to reside outside the fibril core, where it displays conformational and dynamic disorder. **Despite these dynamics it is**, however,

tightly enclosed by the C-terminal PRD domains. **Consequently**, residues in N17 may be solvent accessible, but are nonetheless largely inaccessible to larger macromolecules such as chaperones, kinases, ubiquitinases, and other potential N17 interaction partners. Although in Fig. 6B the known phosphorylation sites T3, S13, and S16 may appear accessible, **Fig. 6A shows** this monomer in context, **underlining** that the N17 segment is fully surrounded by the longer PRD segments. Similarly, the only HttEx1 ubiquitination sites involve lysines K6 **and** K9 in N17, rendering them inaccessible. The buried nature of N17 also rationalizes the low level of engagement by the TRiC chaperone:^{31,57} Although known to bind this part of HttEx1, prior EM studies have shown TRiC to be unable engage HttEx1 fibrils, except near the fibril ends. N17 is also implicated in membrane interactions, such that its preferential exposure at fibril ends may explain the latter to be engaged with the ER membranes **in Fig. 1C**.^{52,58,59}

Revised: In our fibril architecture, the N17 segment is found to reside outside the fibril core, where it displays conformational and dynamic disorder. **Crucially**, however, N17 is tightly enclosed by the C-terminal PRD domains: Residues in N17 may be solvent-accessible, but are nonetheless largely inaccessible to larger macromolecules such as chaperones, kinases, ubiquitinases, and other potential N17 interaction partners, **as can be unambiguously concluded from polymer brush theory (Fig. 8)**. Thus although in Fig. 7B the known phosphorylation sites T3, S13, and S16 may appear accessible, **viewing this monomer in context (Fig. 7A and Fig. 8) underlines** that the N17 segment is fully surrounded by the longer PRD segments. Similarly, the only HttEx1 ubiquitination sites involve lysines K6, K9, **and K15** in N17, rendering them largely inaccessible **in fibrillar HttEx1**. The buried nature of N17 also rationalizes the low level of engagement by the TRiC chaperone:^{33,72} Although known to bind this part of HttEx1, prior EM studies have shown TRiC to be unable **to** engage HttEx1 fibrils, except near the fibril ends. N17 is also implicated in membrane interactions, such that its preferential exposure at fibril ends may explain the latter to be engaged with the ER membranes.^{64,73,74}

Moreover, in part in response to the queries by reviewer 1, we have also expanded our discussion (and data) of the surface features of these polyQ/HttEx1 fibrils and how they are important for HD research. For instance, on pages 2 and 14 we comment on how these surface features are highly relevant to e.g. the design of PET ligands selective for expanded-polyQ inclusions (ongoing research in the HD field; e.g. ref. 10 - Liu et al DOI 10.1021/acs.jmedchem.0c00955). We have expanded the discussion of this point, and mentioned this key topic in more detail in the abstract.

Reviewer #3:

Reviewer #3 (Remarks to the Author):

Integrative determination of the atomic structure of mutant huntingtin exon 1 fibrils from Huntington's disease

The paper provides a meticulous analysis and integration of existing experimental data to develop a structural model of HttEx1 fibrils. The authors used fibrils produced either from recombinant mutant HttEx1 protein or from synthetic peptides as an experimental basis for model generation. The fibril model established here suggests structural information about the polyQ fibril core as well as the relatively flexible N- and C-terminal domains.

In its current form, the study is too preliminary for publication in Nature Communications. However, if the authors are able to address the points raised below, I am happy to review the manuscript again.

Major concerns:

Novelty: The scientific advance in relation to previously reported structural studies (e.g., Nazarov et al., 2022; Boatz et al., 2020 and Matlahov et al., 2022) is too small.

As we noted in response to reviewer 2, above, we strongly disagree with this perspective. None of the prior papers have yielded a defined **atomic** model (which we will make available to the community upon publication via Zenodo and the PDB-Dev database). Key new points include a truly atomistic view of the flanking domains, an accurate model of the polyQ amyloid core, and new models and experimental data of the polyQ fibril surface. The latter is for instance crucial for the design of PET ligands that recognize polyQ fibrils but not other amyloids. (ref. 10; this is an ongoing project in our group, where our own progress required the availability of a detailed atomic-level structural model that is based on, and validated with, experimental structural data).

The reviewer notes three specific prior papers (which were/are all cited in our paper):

- 1) Nazarov et al (10.1021/jacs.2c00509; ref. 22) — this is a cryo-EM study of HttEx1 fibrils, which is focused on explaining HttEx1 fibril polymorphism. As expertly noted by Reviewer 1, the obtained cryo-EM density map had limiting signal/noise, preventing a detailed atomic-level interpretation of the fibril core. The flanking domains were almost entirely invisible. This paper did not produce an atomic-level model.
- 2) Boatz et al (10.1016/j.jmb.2020.06.021; ref. 37) — this is one of our own prior papers, in which we took a step toward understanding the 3D architecture of HttEx1 fibrils. However, that paper lacked any atomic-level modelling, with the only presented models being hand-built to serve as graphical illustrations. No MD was done.
- 3) Matlahov et al (10.1016/j.yjsbx.2022.100077; ref. 38)— this is another paper of ours. The aim of this paper was narrowly focused on the introduction of new ssNMR techniques to probe residues on the fibril surface. The paper did not produce any new structure or model. No MD was done. (The reported techniques represented an earlier, and less-successful effort at resolving the surface glutamines, as a precursor to the new H/D exchange data that we introduce in the revised manuscript)

Thus, none of these three papers included the type of results presented in the current work. The structures presented in this paper are entirely novel to the literature and of great interest to the HD field (indeed, we have already received multiple requests for coordinates since reporting these results on bioRxiv)

Aside from the new structural insights, the manuscript also includes new experimental data, which now include new cutting-edge ssNMR studies of the surface or core of the polyQ core of HttEx1 fibrils by combining HDX and ¹H-detected fast MAS NMR. These new data further boost the novelty of our work compared to prior publications in HD (and the amyloid literature more broadly).

The authors generated two structural models (M1 and M2) for Q44-HttEx1 fibrils through integration of previously reported data and all-atom molecular dynamics (MD) simulations. However, they do not provide specific experimental validation of their model predictions. For instance, it would be important to show how the exchange of glutamines through non-glutamines in the polyQ tract of HttEx1 affects the atomic resolution perspectives. Such experiments would enable verification/falsification of specific predictions.

This reviewer offers an interesting and valuable suggestion for further experimental studies into the structure of polyQ/HttEx1 fibrils, through the introduction of non-glutamine residues within the polyQ segment. Indeed, this approach has already provided valuable insights into polyQ aggregation behavior, both in our own hands and in the work of others. A strategy that we have explored in prior work was to introduce non-Gln residues expected to stabilize or destabilize the formation of β -hairpins by the polyQ segment. For instance, we have studied the impact of L-Pro-Gly and D-Pro-Gly motifs, which are expected to have differential preferences for β -turn formation (see DOI 10.1016/j.jmb.2013.01.016; 10.1073/pnas.1521933113 (especially SI), and 10.1016/j.jmb.2016.12.010). Earlier this year, we built on this work, by starting to incorporate light-responsive azobenzene switches into a polyQ peptide backbone (10.1021/jacs.3c11155), with the specific aim of enabling the kind of studies that the reviewer suggests. However, these are non-trivial - one complicating factor is that mutations or modifications can redirect the complex aggregation pathway of HttEx1 and thus yield unpredictable results that may not correlate only to the final fibril architecture (e.g. based on the impact of mutations on pre-fibrillar structures!)

(NB. Also others have reported interesting work in this direction, but the lack of viable structural models until now have limited the achievable insights from such experiments. Nonetheless, we could mention older work by Ronald Wetzel (ref. 58,60), experiments by Michelle Poirier (PG insertions in a cellular context; ref. 59), work by Randal Halfmann (ref. 61), and several other groups).

In summary, these experiments are non-trivial and beyond the scope of the current work. We have however expanded our discussion of the existence of such studies, noting them in support of our inclusion of a β -hairpin fold in the HttEx1 model (see also our response to the reviewer 1 query on this topic above).

Regarding experimental validation: we have edited our manuscript to better highlight and explain the experimental data that we used to validate our structure (as opposed to the set of experimental constraints used to inform/design our modeling efforts). This is also discussed in responses to other reviewers. We hope that this is now clearer in the revised text, and further supported with the added new data.

The title of the study is somewhat overstated. It suggests that huntingtin exon 1 fibrils prepared from patient tissues have been assessed. The authors have generated a structural model for experimentally generated fibrils. This is clearly an important undertaking. However, huntingtin fibrils formed in vivo may structurally and morphologically be distinct from in vitro generated ones (see also comments below).

We are happy to see here that the reviewer considers the presented work to be “clearly an important undertaking”. We regret any confusion stemming from our title - we did not intend it to be misleading, but were simply trying to create a short-enough title that described the relevance to Huntington’s disease research. We have now slightly revised the title to address this concern:

Original: Integrative determination of the atomic structure of mutant huntingtin exon 1 fibrils from Huntington’s disease

Revised: Integrative determination of the atomic structure of mutant huntingtin exon 1 fibrils implicated in Huntington’s disease

Issues regarding a possibly misleading outlook.

1. Abstract clarification:

I suggest modifying the abstract to explicitly state that the authors incorporated pre-existing EM and ssNMR data. It is important to ensure the language conveys the limited contribution of original experimental data, emphasizing the use of existing data to impose constraints on their model.

In the revised manuscript, we made the distinction between existing and prior data more clear in the abstract (and rest of the text). From reviewer comments, we believe that some new data was assumed to be pre-published, leading to the impression that old data was presented as new. We hope this is clearer now. Moreover, in multiple places we added the phrase ‘in vitro’ to assuage concerns about possible differences being present in cellular aggregates. Finally, we added additional comments about the relevance and meaning of our newly added solid-state NMR data.

Original: Abstract. Neurodegeneration in Huntington’s disease (HD) is accompanied by the aggregation of fragments of the mutant huntingtin protein, a biomarker of disease progression. A particular pathogenic role has been attributed to the aggregation-prone huntingtin exon 1 (HttEx1) fragment, whose polyglutamine (polyQ) segment is expanded. Unlike amyloid fibrils from Parkinson’s and Alzheimer’s diseases, the atomic-level structure of HttEx1 fibrils has remained unknown, limiting diagnostic and treatment efforts. We present and analyze the structure of fibrils formed by polyQ peptides and polyQ-expanded HttEx1. Atomic-resolution perspectives are enabled by an integrative analysis and unrestrained all-atom molecular dynamics (MD) simulations incorporating experimental data from electron microscopy (EM), solid-state NMR, and other techniques. Visualizing the HttEx1 subdomains in atomic detail helps explaining the biological properties of these protein aggregates, as well as paves the way for targeting them for detection and degradation.

Revised: Abstract. Neurodegeneration in Huntington’s disease (HD) is accompanied by the aggregation of fragments of the mutant huntingtin protein, a biomarker of disease progression. A particular pathogenic role has been attributed to the aggregation-prone huntingtin exon 1 (HttEx1) fragment, whose polyglutamine (polyQ) segment is expanded. Unlike amyloid fibrils from Parkinson’s and Alzheimer’s diseases, the atomic-level structure of HttEx1 fibrils has remained unknown, limiting diagnostic and treatment efforts. We present and analyze the structure of fibrils formed by polyQ peptides and polyQ-expanded HttEx1 in vitro. Atomic-resolution perspectives are enabled by an integrative analysis and unrestrained all-atom molecular dynamics (MD) simulations incorporating experimental data from electron microscopy (EM), solid-state NMR, and other techniques. Alongside the use of prior data, we report new magic angle spinning NMR studies of glutamine residues of the polyQ fibril core and surface, distinguished via hydrogen-deuterium exchange (HDX). Our study provides a new understanding

of the structure of the core as well as surface of aggregated HttEx1, including the fuzzy coat and polyQ-water interface. The obtained data are discussed in context of their implications for understanding the detection of such aggregates (diagnostics) as well as known biological properties of the fibrils.

2. Section and Figure 1 clarification:

- I recommend moving Figure 1, partly or in total, to supplementary material as it summarizes findings from other authors and does not contain original results.
- I suggest removing Cryo-ET data (panels B and C) from Figure 1 due to its lack of contribution to the model and the potential to mislead regarding in vivo representation. Furthermore, the authors do not take any data from pre-existing cryo-ET experiments to build their structural model.

We decided to substantially change Figure 1, while also making more clear which data were reused from prior work. As suggested, we completely removed the tomography images (reused from a prior paper) that this reviewer considered overly suggestive of ex-vivo or cellular analysis. Because reviewer 1 specifically requested a discussion of prior data, we decided not to remove prior ssNMR results completely, but rather to move those to a new SI figure (Supplementary Figure 1).

Issues regarding overall assumptions

1. Avoiding to imply in-cell application:

- I recommend removing the in-cell aggregate data from Figure 1 and refraining from making assumptions about in-cell application to align with recent evidence regarding other amyloid fibrils. There is increasing evidence that the structural features of fibrils formed in vitro by proteins related to neurodegenerative disorders have limited resemblance to the structure of fibrils found in human patients.

We have expanded an explicit discussion of the possibility that aggregates in cells or patients would be different from those formed in vitro (in the Discussion part of the paper; bottom of page 15). In this section we provide our preliminary rationale suggesting that the currently available data on polyQ fibrils may suggest a limited ability to form qualitatively different fibril core structures, but naturally this remains to be tested experimentally.

Relevant text from pages 15-16:

“Naturally—although we expect the obtained fibril structure to be a good representation of the fibrils present in our samples and also to have strong predictive and descriptive qualities for cellular HttEx1 aggregates—it cannot be excluded that cellular factors (ranging from chaperones to membrane interactions) may modulate the aggregation mechanism to the extent that the mature fibril architecture differs from the one obtained in vitro. Yet, unlike for other amyloid proteins, a remarkable feature of the HttEx1 ssNMR studies is that multiple groups have studied a variety of HttEx1 fibrils and always found the same signature spectra that are connected to the structural parameters used to construct our model^{19–21,31,37}. Indeed, there is little evidence for a qualitative change in fibril architecture. Thus, we are inclined to expect that while cellular conditions may change certain details in the fibril structure, they would not fundamentally change the protofilament architecture.”

- For that reason, I recommend avoiding speculations like the one found in line 248, in which authors compare cellular aggregates with in vitro aggregates.

We have also removed the sentence at line 248, as suggested. We do note that this sentence simply repeated statements in prior cryo-ET papers that suggested similarity between fibrils formed in cells and in vitro.

Removed text: Cryogenic electron tomography analysis of cellular httEx1 aggregates have indicated that the fibrils may be structurally similar to those formed by purified proteins,¹⁵ which would suggest that our model also applies to cellular inclusions.

Issues regarding the MDS methodology

1. Explanation of forcefield choice:

- I suggest explaining the choice of forcefields and reasons behind the discrepancy observed with the CHARMM36m.

We have clarified our choice of force fields, stating now explicitly that they represent each of the three main force field families:

Original: Acknowledging the limitations of a classical mechanics approximation (force field) of an inherently quantum system, we employed three state-of-the-art MD force fields **in parallel**: AMBER14SB,⁷⁵ OPLSAA/M,⁷⁶ and CHARMM36m.⁷⁷

Revised: Acknowledging the limitations of a classical mechanics approximation (force field) of an inherently quantum system, we employed **in parallel** three state-of-the-art MD force fields **(one from each of the main force field families)**: AMBER14SB,⁴² OPLSAA/M,⁴³ and CHARMM36m.⁴⁴

The particular parametrization details that lead CHARMM36m in this specific case to behave differently from AMBER14SB and OPLSAA/M are in our opinion—while certainly interesting to the biomolecular force field development experts—beyond the scope of the current investigation. That said, we do find it important to clarify to the non-expert reader that classical force fields are approximations, and different approximations can lead to different outcomes, which therefore should be tested against robust experimental data. We believe that by showing the differing behaviour of CHARMM36m, and by validating the results of OPLSAA/M and AMBER14SB against the ssNMR-derived dihedral angle limits, we achieve this goal.

2. Replicate number explanation:

- Did the authors carry out replicates of their simulations? I recommend including a section explaining the rationale behind the chosen number of replicates or the lack of them.

For studies testing the structural stability of biomacromolecular conformations, truly independent replicates are challenging to create. The typical approach is to start from N conformationally **identical** replicates, with differing initial atom velocities. Unfortunately, such N simulations will

not be particularly independent, as the emerging phase-space trajectory of a folded (or, as in the present case, misfolded) biomacromolecular conformation is strongly determined by its atom positions, and much less so by their momenta. Consequently, to obtain N meaningful replicates, one has to create N **independent** sets of initial atom positions—but each set must still be representative of the (mis)folded conformation to be tested.

That said, we fully agree with the reviewer’s implication on the principal importance of studying independent replicates. To this end, we tried our best to create for each of the 30 polyQ core candidates a second, independent, set of initial atom positions still representative of the conformation to be tested. Our approach was to perform the initial restrained-relaxation steps by restraining the dihedral angles (instead of the heavy-atom positions as before). The outcome of these independent replicates is the same as before: M1 and M2 are stable in AMBER14SB and OPLSAA/M, in CHARMM36m all 30 candidate structures are unstable (Supplementary Fig. S4).

3. Addition of a Limitations section:

- I suggest including a section explicitly stating the limitations of the investigations in this study.

In accordance with this suggestion, we have added a new section to the revised manuscript, entitled “Caveats and fibril polymorphism” (section 2.7; page 14). In this section we brought together a number of caveats that were previously in this parts of the manuscript, while also adding a few additional new considerations. We also discuss in some detail how polyQ protein fibrils are highly heterogeneous, and that any MD model would only be able to capture the main shared aspects of the fibril structures.

4. Figure 2 Panel D addition:

- I recommend adding a figure to display a couple of “failed” models and discussing why they showed low stability compare to M1 and M2.

Following the reviewer’s suggestion, we added supplementary Fig. S6: Scatter plots displaying the χ_1 and χ_3 dihedral angles (similar to Fig 2D) in the last MD simulation frame for all the 30 candidate models (in gray), as well as the initial values of these dihedrals (in color). We also revised the text in Results to include the insight provided by this new supplementary figure on why the “failed” models showed low stability compared to M1 and M2:

Original: Notably, these two stable 3D lattices (called M1 and M2) nicely capture known features of polyQ amyloid structure. For instance, the experimental constraints offered by ssNMR dihedral angle measurements are open to more than one interpretation, as is typical for such experiments.^{5,42} Previously, this ambiguity and angular uncertainty yielded qualitative models useful for illustrative purposes, but not robust upon close inspection. Here, we have narrowed down the viable and physically plausible models to only two, with these models identifying specific narrow regions within the equivocal dihedral angle space (Fig. 2D). In both models, the antiparallel β -sheet harbors strand-specific Gln structures occupying the side chain rotamers known as pt20° and mt-30°, according to Ref. 43.

Revised: Notably, these two stable 3D lattices (called M1 and M2) nicely capture known features of polyQ amyloid structure. For instance, the experimental constraints offered by ssNMR dihedral angle measurements are open to more than one interpretation, as is typical for such experiments.^{8,47} Previously, this ambiguity and angular uncertainty yielded qualitative models useful for illustrative purposes, but not robust upon close inspection. Here, we have narrowed down the viable and physically plausible models to only two, with these models identifying specific narrow regions within the equivocal dihedral angle space (Fig. 2D). **Interestingly, unlike**

all the 28 unstable models, **M1** and **M2** display no Gln residues with $\chi_3 \approx 180^\circ$ (Supplementary Fig. S6); rather, in both models, the antiparallel β -sheet harbors strand-specific Gln structures occupying the side chain rotamers known as pt20° and mt-30°, as defined by to Ref. 48.

5. Solvation of polyQ core simulations:

- I propose clarifying whether the polyQ core simulations were performed solvated or in vacuo. In case they were performed in vacuo, please provide references for the methodology where it states that the selected forcefields are suitable for those type of simulations.

The polyQ core simulations were performed on an infinite periodic lattice of the 8-Gln unit cells, not in vacuo. We have clarified this setup in the Results:

Original: After applying this additional filtering, 30 distinct unit cell architectures still remained possible. To investigate the structural stability of these 30 models, we carried out all-atom MD simulations in a fully periodic setting on them; see Methods for details.

Revised: After applying this additional filtering, 30 distinct unit cell architectures still remained possible. To investigate the structural stability of these 30 models, we carried out all-atom MD simulations in a fully periodic “infinite-core” setting on them; see Methods for details.

As well as in the Methods:

Original: For each of these 30 unit cells candidates, we performed all-atom MD simulations. The fully periodic MD simulation box contained 40 identical unit cells, that is, a total of $8 \times 40 = 320$ Gln residues, organized as a stack (along y) of 4 antiparallel β -sheets, with each sheet comprising 8 β -strands of periodic (along x) Q_{10} peptides.

Revised: For each of these 30 unit cells candidates, we performed all-atom MD simulations. The fully periodic MD simulation box was filled by 40 identical unit cells, that is, a total of $8 \times 40 = 320$ Gln residues, organized as a stack (along y) of 4 antiparallel β -sheets, with each sheet comprising 8 β -strands of periodic (along x) Q_{10} peptides.

Experimental data clarifications

1. HETCOR experiment:

I suggest clarifying whether HETCOR experiments were performed using HttQ44Ex1 or the peptide D2Q15K2.

This comment refers to what is now Figure 2G of the revised manuscript. Note that this part of the figure has been changed and updated compared to the original submission, also to address comments from other reviewers. In the revised caption we provide a better description of the sample, and we have added a new Supplementary figure S7 to compare spectra of protein and peptide fibrils. The HETCOR experiments in the original paper were done only for the synthetic peptide, but we have now added comparable NMR data obtained on the polyQ core of HttEx1 protein fibrils. The latter data are now in Fig. 2G and both samples are compared in the SI Figure S7. We study both types of fibrils in order to test that the detected signals are really from the glutamines: in the synthetic peptides only two glutamines were outfitted with isotope enrichment such that clean data on the polyQ core could be obtained. We also expanded the Methods section with more information on the employed peptide and protein fibril samples in the prior and newly added NMR samples.

2. References:

Appropriate references for experimental results described in line 96 are missing.

This comment refers to prior reports on the topic of the a/b conformer each forming separate beta-strands, which is also discussed in response to reviewer 2 comments above. We added the missing citations to the affected line (page 4 in revised text). We also slightly expanded this point, given the questions of reviewer 2 above, and refer readers to the new Supplementary Figure S1. The revised line now reads:

“A key additional consideration is that ssNMR has unambiguously shown sequential residues within each β -strand to have the same backbone conformations (based on identical chemical shifts; Supplementary Figure S1B).^{8,21,35}

3. Figures 3 and 5:

I suggest clarifying the origin of TEM images in Figure 3 and whether the data in Figures 5B and 5D are original or taken from other authors (as can be inferred by the text).

The TEM data in Figure 3 and the 2D ssNMR (TOBSY) spectrum in Figure 6D (Fig. 5D in the original text) were new data, not previously published. It is possible that citations to similar prior studies (by both ourselves and other groups) in the text gave the impression of data reuse. We have tried to clarify the affected text.

Related to Figure 3, we rephrased the text as follows, avoiding potentially confusing citations to prior EM studies on similar samples (page 7)::

“..we constructed $D_2Q_{15}K_2$ fibrils comprising seven anti-parallel β -sheets, consistent with the 5.5- to 6.5-nm fibril width seen experimentally by TEM (Fig. 3A).”

Related to Figure 6D, we rephrased the text as follows; new text (page 11; near line 295):

“The latter is evidenced by those residues showing up in INEPT-based MAS NMR measurements that are selective for highly flexible residues, such as the 2D INEPT-TOBSY spectrum in Fig. 6D. These data are consistent with prior studies using similar methods as well as relaxation measurements to show the high flexibility of the C-terminal tail.”

The chemical shift data in Fig. 6B (previously Figure 5D) are indeed representing previously published data. We have tried to make this clear in the revised figure caption by adding the phrase “... replotted from previously reported work.” along with the appropriate citation (see caption of Figure 6).

4. Line 241:

I recommend clarifying or removing the comparison between the model and phase transitions observed in the nuclear pore complex that appears in line 241.

The mentioned sentence in section 2.6 has been removed (near line 300 in revised text).

In conclusion, we would like to thank the reviewers for their appreciation for our work and their insightful and constructive comments that have allowed us to significantly improve the manuscript. We hope and believe that our responses and revisions fully address all questions raised and make the paper ready for publication.

Responses to Reviewer Comments

We thank all reviewers for their insightful comments and suggestions. Below we reproduce the comments verbatim and provide a point-by-point response to each query, along with specific information about accompanying changes in the revised manuscript.

Reviewer #1:

Reviewer #1 (Remarks to the Author):

The authors have made major revisions to address my criticisms of the original version of this manuscript. They have clarified and expanded important parts of the text, added new data, and modified the figures. I am fully satisfied with these revisions. I have no further requests for revisions.

We thank the reviewer for their positive response and appreciation of our revised manuscript.

Reviewer #2:

Reviewer #2 (Remarks to the Author):

The authors carefully addressed all the reviewers' comments. As a result, the manuscript was significantly improved.

We thank the reviewer for their appreciation of our revised manuscript.

However, I am still not fully convinced of the novelty of the results presented that would justify publication in Nature Commun.

My main concerns are:

Previous data/knowledge from the same group: in 2016, part of the same group of authors had already published the existence of the “a” and “b” conformers for the side chains in the paper by Hoops et al. They had also shown that “a” and “b” must be adjacent in a beta-sheet. I understand that further NMR experiments were performed here to corroborate these earlier results and provide data on the fibril-water surface (see my comment below).

We respectfully stand by our previous arguments regarding the novelty and importance of our results. Until now, no robust and comprehensive atomic-level structures of either polyQ peptide aggregates or of HTTex1 fibrils exist in the literature. Previous work has only shed light on certain details, and/or yielded highly qualitative or schematic models. We expect substantial impact of the newly available atomic-level insight into HTTex1 fibrils, especially given a newly realized central role of this protein construct in HD pathogenesis, in the last few years. (The expected impact is enhanced by the revisions that we made in the current and prior revisions, for which we thank the reviewers.)

“a” and “b” conformers: MD simulations were used to create many structural models for these conformations. For this purpose, many simulations were performed, which led to the conclusion that only two of them are stable (referred to as “M1” and “M2” – please changed “devoted” to “denoted” to l. 123

on p. 5). However, if you look at Fig. S2, you can see - without running MD simulations (I'm a simulation scientist, so I have nothing against MD!) - that most of these conformations are not stable because no H-bonds can be formed between the side chains in these conformations.

We thank the reviewer for noting the typo (on line 123 of page 5), and have corrected it:

Original: these two stable 3D lattices (de**v**oted M1 and M2) capture

Revised: these two stable 3D lattices (de**n**oted M1 and M2) capture

We also thank the reviewer for drawing our attention to the fact that the visualization in Fig. S2 indeed appeared confusing. We had created it just to introduce the concept of (χ_1 , χ_3) dihedral angle rotations to the reader; thus, for visualization, a single structure was being modified by hand. Although this nicely (we thought) showed how the 8 structures are related, it did lead to the confusion that some side-chain-side-chain H-bonds seemed absent. We have, therefore, now replaced all the 8 snapshots in Fig. S2 with their dihedral-restraint-equilibrated versions, in which the H-bonds indeed are present. We think that this is clearer for the reader.

We have also added the following clarification to the caption of Fig. S2:

Revised: The shown snapshots of these eight unique pairs are taken from the dihedral- restraint-equilibrated conformations, where the dihedral angles (χ_1 , χ_3) are restrained to their initial values (listed next to the corresponding sidechain, see also Supplementary Fig. S3).

Why did the authors not bother to provide the χ_1 and χ_3 values in Fig. S2 so that the reader can easily relate the information in Fig. S2, Fig. S3 and the main manuscript?

We now provide the (χ_1 , χ_3) dihedral angle pair values next to each side chain in Fig. S2.

Another interesting question, in my opinion, would have been why the alternation of “a” and “b” conformers in a beta-strand (left figure in Fig. 1H) is not possible. I understand that this information was obtained from NMR, but MD would have provided the opportunity to give an explanation.

We agree that there are many interesting questions that could be further investigated building on our here-presented structural analysis. This includes the interesting point that the reviewer raises here. That said, we do consider those questions better suited for future studies.

It is nice that Amber14SB and OPLSAA/M match, but Charmm36m is also a very good force field, also for the side chains. So it is a mystery to me why the performance of Charmm36m is exceptionally poor here. The authors should have checked what is happening here.

While we again fully agree with the reviewer that the difference observed between Amber and OPLS compared to Charmm36 is of relevance, especially for future force field development, we do feel that such detailed parameter analysis is also a topic for follow-up work.

By the way, it is customary not to show the equilibration results that contain constraints, as was done here. This would have avoided misunderstandings with reviewer 3.

We agree with the reviewer that it is common to show data only after restrained equilibration. However, as some of the 30 candidates proved unstable already during the (position) restraint equilibration, we decided it to be less confusing to a non-expert reader if we show also the restrained equilibration (otherwise it would appear that some candidates had lost stability before any time steps were taken).

Another side comment: polyQ prefers antiparallel beta-sheets - which is emphasized by the authors on p. 5 - as this is the more stable beta-sheet compared to parallel alignment. And since all residues are Q, they can adopt this structure, whereas in other amyloids this would lead to many energetic mismatches between amino acids; hence the parallel orientation is adopted.

We thank the reviewer for this comment, on one of the many interesting features of these unusual protein aggregates. We agree with this assessment, which we expect to accurately describe some of the molecular underpinnings of why polyQ aggregates differ in this fundamental way from more 'canonical' amyloids.

Surface-water interactions: It is not surprising that the conformations of the side chains are less ordered, since the side chains form H-bonds with each other in the fibril core and compete with H-bonds between water and the side chains at the surface. Thus, it is not a surprising finding; nonetheless, it should be reported as done by the authors.

We thank the reviewer for supporting the way we reported our data. Indeed, we hope to have conveyed that the water-exposed side chains are less ordered than the side chains tucked in the fibril core, as anticipated by the reviewer. However, we also note that they still "do not show fully free side-chain motion and retain some of the structural features that determine the molecular profile of the corrugated polyQ amyloid core surface." (lines 198–199 on page 8)

In summary, I stand by my earlier assessment that the work was carefully done and should be published, but I don't see the major breakthrough that would justify publication in Nature Commun. However, if the other reviewers see it differently, I am happy to be proved wrong. The strength of the paper is that it brings together information from previous studies, supplements this with additional NMR data and provides structural models based on MD data. The MD data was very carefully prepared, but the structural models could have been derived with less effort.

We appreciate the reviewer's recognition of the care taken in our work. However, we respectfully disagree with the suggestion that the models could have been obtained with less effort. In our view, shortcuts could have compromised the reliability and robustness of our results, which would then limit their usefulness to the HD and amyloid research fields.

Reviewer #3:

Reviewer #3 (Remarks to the Author):

The manuscript was significantly improved and is now ready for publication in Nature Communications.

We thank the reviewer for the positive comments on the revised paper and its suitability to the journal.

I have, however, two minor points that should be considered by the authors to further improve this manuscript:

1) The authors use the abbreviation Httex1 in their manuscript. However, the term Htt is utilized to indicate mouse huntingtin. Please see the recently reported nomenclature for HD research (DOI: 10.3233/JHD-240044). I assume that human HTT exon 1 fragments were exclusively analyzed in this study. Thus, the term HTTex1 (or HTTEx1) should be used instead of HttEx1 in this manuscript.

We thank the reviewer for the suggestions on the HD/HTT nomenclature and for bringing the new report to our attention. We have opted to use HTTex1 as recommended by the reviewer and modified the text and figures correspondingly.

2) Figure 8: in D black and gray colors were used to indicate HTTex1 fibrils. I do not fully understand what the gray coloring is good for. May be for clarity reasons coloring should be changed in Fig.8.

The gray coloring was meant to indicate the 3D-character of the fibril. The figure shows two subsequent layers of protein (in the fibril), which were colored differently: black or gray, respectively. The reviewer comments made it clear that this was not clear in the original caption and figure. We have clarified the visualization by revising the caption of Fig. 8:

Original: Silhouette of the HTTex1 fibril (two consecutive layers of monomers along the fibril growth direction shown in black and gray), highlighting its post-translational modification sites in colour:

Revised: Silhouette of the HTTex1 fibril. The view is along the fibril growth direction. Shown are two consecutive monomer layers: the top layer in black, the layer behind it in gray. Post-translational modification sites in the top layer highlighted in colour:

We thank the reviewers for their constructive comments and we think that the revised manuscript is improved and clearer. We hope that these revisions make it more clearly understandable to a broader audience.